# ProtoKV: Streaming Video Understanding under Delayed Query with Summary-State Memory

**Le Tu Ngoc Minh** [* 1]  **Jinyeong Lim** [* 1]  **Dongsu Han** [1 2]

## Abstract

Streaming video understanding (SVU) must answer queries that arrive asynchronously while visual tokens stream continuously under strict GPU-memory and query-time latency budgets. A key challenge is delayed query: decisive cues may appear briefly, yet many subsequent updates occur before the query arrives, increasing the risk that those cues are evicted or diluted under bounded memory. We propose ProtoKV, a constant-footprint SVU memory that represents far history as a fixed-capacity summary state rather than retaining token instances. ProtoKV keeps an exact near-window KV cache and aggregates older content into a semantic–spatial prototype bank with residual statistics. At query time, each prototype is exposed through a bounded pseudo-token interface that is drop-in compatible with standard attention. Under matched budgets and comparable query-time cost, ProtoKV improves accuracy by up to 12.5 points over token-retention baselines on SVU benchmarks in the long-delay regime, with gains that grow as query delay increases.

## 1. Introduction

Streaming video understanding (SVU) systems must reason over continuous video signals in real-time (Zhang et al., 2025; Niu et al., 2025; Lin et al., 2024). Despite progress on offline benchmarks, real-world deployment requires handling continuous frames and asynchronous queries under a fixed GPU memory budget (Chatterjee et al., 2025). In this setting, the model cannot assume access to the full clip nor condition computation on a known query; instead, it must maintain an evolving online state that remains immediately query-able upon demand (Chen et al., 2024; Zhang et al., 2025).

A central difficulty in SVU is delayed query, where decisive cues (e.g., a momentary state change) appear briefly and long before a query is issued (Yang et al., 2023; Grauman et al., 2022). Here, delay is not only a time gap but also the amount of post-evidence update pressure accumulated before answering. As the temporal gap between evidence and query grows, answering quality depends increasingly on what the system preserves from far history; this induces a near–far regime where recent content is kept exactly within a short window, while older history must rely on a compact representation.

Bounded-memory designs typically occupy two extremes: sliding-window baselines maintain constant memory by retaining only recent content but inevitably lose evidence once it expires (Xiao et al., 2024; Beltagy et al., 2020). Conversely, offloading or retrieval systems recover fine-grained details but add significant query-time overhead and variance (Di et al., 2025; Sun et al., 2025). Between these extremes lies a critical operating point for interactive systems: always-on; query-agnostic maintenance; a fixed on-device footprint usable directly at query time without reconstructing the past.

Recent work advances this operating point via online KV-cache compression under a hard memory cap for SVU (Kim et al., 2025). While this class of methods yields predictable query-time cost, online retention decisions are brittle when answers depend on rare but decisive evidence across long delays. More broadly, tying far-history to specific token instances means that once a token is dropped, its associated fine-grained cue is lost (Zhang et al., 2023; Li et al., 2024a).

We propose ProtoKV, a constant-footprint SVU memory mechanism in this regime that shifts far-history representation away from token retention via a two-tier online state: (i) a short near window storing exact KV for recent frames, and (ii) a fixed-capacity far memory summarizing older history into object-centric prototypes with lightweight residual statistics. ProtoKV does not retain token-level KV for the distant past; instead, it leverages pseudo-tokens and mass-aware weighting to allow stan-

---
[*]Equal contribution  [1]School of Electrical Engineering, KAIST, Daejeon, Republic of Korea  [2]Kim Jaechul Graduate School of AI, KAIST, Daejeon, Republic of Korea. Correspondence to: Dongsu Han <dhan.ee@kaist.ac.kr>.

*Proceedings of the 43rd International Conference on Machine Learning*, Seoul, South Korea. PMLR 306, 2026. Copyright 2026 by the author(s).

dard attention to access far-history evidence under a constant query-time footprint. This design is tailored to delayed queries depending on past event occurrence rather than instantaneous visual content. Evaluations across four streaming video benchmarks demonstrate that ProtoKV improves accuracy and degrades more gracefully than token-retention baselines as evidence becomes increasingly distant under matched memory budgets. Our code is available at https://github.com/kaist-ina/ProtoKV.

**Contributions.** (1) We introduce a far-memory mechanism for SVU that summarizes distant history into a fixed-capacity, spatiotemporally consistent prototype bank augmented with residual statistics. (2) We propose pseudo-token synthesis with mass-aware weighting, enabling standard attention to exploit far-history evidence under a bounded query-time footprint. (3) We introduce a delayed-query protocol with retrospective-query filtering to keep invariant ground truth.

## 2. Background and Problem Setup

### 2.1. KV Growth as an Operational Constraint in SVU

Streaming video understanding (SVU) requires maintaining an online state to process continuous video streams and answer arbitrary queries without replaying past frames (Chen et al., 2024; Chatterjee et al., 2025). In Transformer-based VLM, this state scales linearly with time as each visual token is appended to the KV cache (Dosovitskiy et al., 2021). Given the unbounded nature of streaming, this context growth inevitably exceeds fixed deployment budgets, creating a fundamental scalability bottleneck (Kwon et al., 2023; Zhang et al., 2023).

This pressure is amplified in video due to the high token arrival rate, quickly producing large token histories even with aggressive sampling. As a result, SVU systems must maintain an online state that supports immediate answering at query time, but whose memory cost does not scale with stream length. While simple heuristics like window truncation or token downsampling can stabilize memory (Xiao et al., 2024; Beltagy et al., 2020; Li et al., 2024b), they do so by sacrificing long-range evidence or fine-grained details, respectively. Consequently, SVU demands bounded-memory mechanisms that preserve past representations in a fixed-capacity footprint while remaining compatible with standard attention at query time.

### 2.2. Bounded-Memory SVU and Prior Approaches

In contrast to offline video understanding, where the full sequence is accessible and models can compress or revisit tokens with complete context (Zhou et al., 2025; Fu et al., 2025), SVU must process frames online without knowing the stream horizon in advance. In practice, this makes it

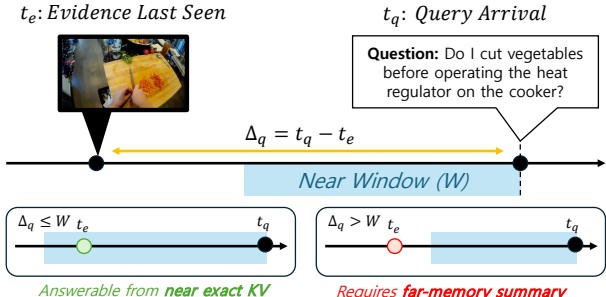

*Figure 1.* Query delay in streaming video understanding. When the delay exceeds the near window, answering requires information preserved in far memory.

unrealistic to "buffer the whole video" before answering. We therefore focus on bounded-memory SVU under three deployment-driven constraints. First, the memory footprint must be capped by design (constant or hard-bounded). Second, the online state must be maintained query-agnostically: the system cannot tailor what it stores to a future query it has not seen yet, nor can it replay the stream at query time. Third, the stored state must be immediately consumable at query time without expensive reconstruction steps that introduce latency variance.

While some works utilize end-to-end trained compression (Chatterjee et al., 2025), we focus on training-free, drop-in mechanisms that are easier to deploy across models and backbones. Prior approaches span a spectrum of trade-offs under these constraints. Sliding-window KV caching (SWA) guarantees constant memory by retaining only the most recent window, yielding predictable cost but losing evidence once it exits the window (Xiao et al., 2024; Beltagy et al., 2020). Online token-retention/compression methods (Kim et al., 2025) maintain a fixed KV budget by selecting, merging, quantizing, or evicting token instances over time. While they can preserve some far history, they still represent far evidence as a subset of token instances, and must continually decide which instances survive under a hard cap. Offloading/retrieval-based SVU (Di et al., 2025) stores history in slower tiers or retrieves it at query time to recover fine-grained details; however, the resulting query-time overhead and variance can be a practical weakness when responsiveness is central (e.g., storage/network contention or strict tail-latency targets). These trade-offs shape the operating point required for practical SVU deployments: always-on, query-agnostic maintenance of a constant-footprint on-device state that can be directly used at query time, without reconstructing the past.

### 2.3. Query Delay and Post-Evidence Update Pressure

We now formalize query delay as the key stressor in SVU. A decisive cue may appear briefly, yet the system must con-

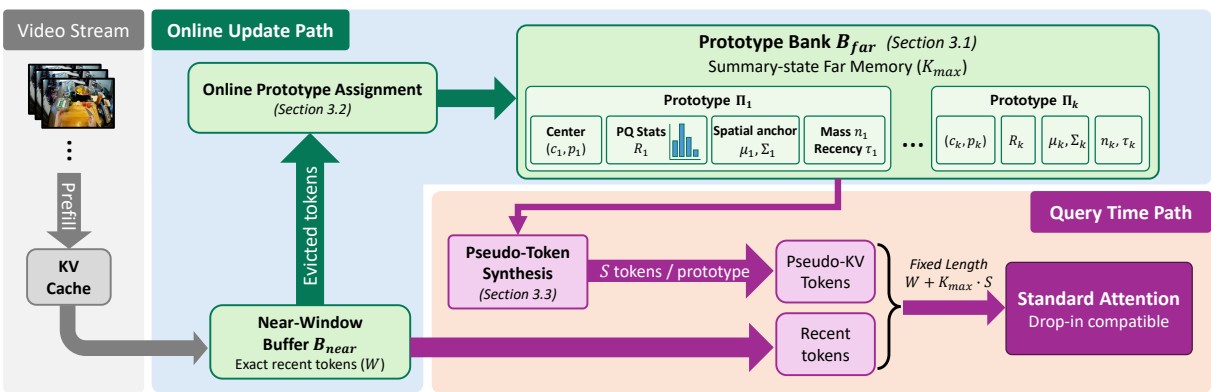

*Figure 2.* ProtoKV maintains an exact near-window KV buffer for recent tokens and, as the video stream continues, continuously aggregates evicted history into a fixed-capacity, object-centric prototype bank via online assignment. At query time, it synthesizes a bounded number of pseudo-KV tokens per prototype and concatenates them with the near-window tokens, yielding a fixed-length sequence that can be consumed by standard attention without architectural changes.

tinue ingesting frames and updating a bounded state until the query arrives. During this interval, the cue can be either explicitly removed or gradually degraded by repeated eviction/compression decisions under a fixed footprint. Fig. 1 depicts this setting, where evidence occurs at time $t_e$ and the query arrives later at $t_q$ after many intervening updates. In an idealized view, for each query $q$, let $t_e$ denote the last-seen timestamp of the decisive evidence, and $t_q$ denote the query arrival time. We define the query delay as $\Delta_q = t_q - t_e$. Larger $\Delta_q$ means the system must preserve query-relevant information across more intervening streaming updates.

In practice, SVU benchmarks (Zhang et al., 2025; Niu et al., 2025; Lin et al., 2024) typically do not provide per-query evidence timestamps $t_e$. Thus, our delayed-query evaluation uses the dataset-defined query timestamp $t_0$ as a reference and shifts the query to $t_q = t_0 + \Delta$ (used as an empirical proxy for $\Delta_q$), while restricting to retrospective query types with invariant ground-truth.

Importantly, $\Delta$ is not merely a time gap; it serves as a proxy for the *amount of post-evidence update pressure* accumulated before answering. During this interval, the system must repeatedly (i) encode incoming frames and (ii) update its bounded internal state. Because these updates are query-agnostic, the system must commit to maintenance decisions before knowing which cues will later be queried, and thus cannot selectively preserve only query-useful evidence. Under any bounded-memory mechanism, far history undergoes many potentially lossy operations such as eviction, quantization, merging, or summarization, whose errors can compound with continued updates. This perspective naturally induces a *near–far regime*: a recent near window can be kept at high fidelity, but once evidence falls outside that window, the model must rely on a more compressed far representation. Accordingly, we often observe a sharp

degradation around the near→far transition, followed by a slower decline as $\Delta$ increases further.

A common strategy for bounded memory is *token retention*, which preserves far history by keeping selected token-level KV instances (Zhang et al., 2023; Li et al., 2024a; Kim et al., 2025). However, under large $\Delta$, token retention faces intrinsic challenges. First, retained tokens must compete for a fixed capacity over a long stream; early retention choices can become suboptimal as new content arrives, forcing repeated replacement and yielding compounded information loss. Second, retaining token instances does not directly control whether the remaining set stays *query-usable* after many updates: the mechanism may preserve many redundant or weakly relevant instances while dropping rare but decisive cues, causing abrupt failures once the evidence leaves the near window. These observations motivate representing far history as a *fixed-capacity summary state* rather than as a set of retained token instances, aiming for more graceful degradation as evidence becomes increasingly distant.

## 3. ProtoKV

We present **ProtoKV**, a constant-footprint KV memory for streaming video LLMs under arbitrary query arrivals. The central challenge is *post-evidence update pressure*: after decisive evidence appears, the system must absorb many subsequent updates before the query arrives, and token-instance retention can become increasingly brittle under a fixed budget as this delay grows.

ProtoKV maintains a bounded online state with (i) a footprint independent of stream length, (ii) query-agnostic online maintenance, and (iii) drop-in query-time consumption via standard attention. It uses a *two-tier design*: an exact near-window KV cache of length $W$, and a token-instance-free far summary state implemented as a prototype bank

of capacity $K_{\max}$. As tokens leave the near window, they are absorbed online into the bank via a *continuity-aware assignment rule*, allowing far memory to accumulate evidence rather than repeatedly overwriting it.

At query time, each prototype is exposed as $S$ pseudo-KV tokens synthesized from its centers and lightweight residual statistics, with mass-aware weighting to reflect token multiplicity. Concatenating near exact KV with far pseudo tokens yields a bounded context length $L_{context} = W + K_{\max} \cdot S$, keeping query-time cost bounded even when decisive evidence is primarily supported by far history. Unlike token-retention schemes that must decide which instances to keep under a fixed budget, we maintain a persistent summary state whose updates refine rather than replace evidence.

### 3.1. State representation and prototype bank

**Near exact state.** We maintain a ring buffer $B_{\mathrm{near}}$ that stores the most recent $W$ tokens' exact KV pairs. This near window preserves fine-grained, temporally local evidence without approximation, serving as an exact anchor when relevant evidence lies within the recent horizon.

**Far summary state.** For older history, ProtoKV maintains a fixed-capacity prototype bank $B_{\mathrm{far}} = \{\Pi_k\}_{k=1}^{K_{\max}}$, where each entry $\Pi_k$ is an aggregated summary of many evicted tokens and stores no per-token KV pairs for far history (Rae et al., 2020). Each prototype stores only constant-size sufficient statistics: representative centers $(c_k, p_k)$, a mass $n_k$, residual statistics $R_k$, and optional spatiotemporal metadata $(\mu_k, \Sigma_k, \tau_k)$ used for prototype assignment. Consequently, far-memory cost scales with $K_{\max}$[1], not with the number of processed tokens. Large $\Delta$ stresses far memory primarily via post-evidence update pressure, not mere 'oldness'. Persistent prototype-level state reduces churn (overwrite/switching), allowing sparse cues to remain represented across many subsequent updates.

**Initialization.** As long as empty prototype slots remain, evicted tokens are routed to a new slot rather than absorbed into an active prototype. This prevents premature merging of semantically distinct events under a partially-filled bank and ensures the full $K_{max}$ capacity is utilized as the stream progresses. Empty prototype slots are initialized upon first assignment by setting the centers from the incoming token and resetting the associated statistics.

### 3.2. Online update: eviction, assignment, and prototype maintenance

As the stream progresses, the KV cache of each new token is appended to $B_{\mathrm{near}}$. When $|B_{\mathrm{near}}| > W$, the KV pair of

---

[1]We keep $K_{max}$ fixed (independent of stream length) under each memory budget; thus update-time complexity is constant with respect to stream length.

the oldest token is evicted and absorbed into $B_{\mathrm{far}}$. A key difficulty is that far memory is updated continuously after a decisive cue occurs; without a continuity-aware rule, evicted tokens from the same underlying object/track can be scattered across prototypes, causing prototype switching and fragmentation under sustained update pressure. To address this, we assign evicted tokens using a continuity-aware objective that jointly considers key similarity, spatial consistency, and recency, and then update the selected prototype as a persistent summary state.

**Prototype assignment.** For an evicted token $i$ and KV cache $(K_i, V_i)$, let $s_i = (x_i, y_i)$ denote its normalized 2D spatial coordinate in the frame. We assign it to the prototype minimizing the online cost:

$$k^\star = \arg \min_k \Big[ -\cos(K_i, c_k) + \lambda_{\mathrm{sp}} d_{\mathrm{Mah}}(s_i; \mu_k, \Sigma_k) \\ + \lambda_{\mathrm{idle}} \mathbf{1}[idle(\tau_k)] \Big] \quad (1)$$

The objective combines three terms. (i) *Key-space similarity*: $-\cos(K_i, c_k)$ groups tokens whose keys are close to the prototype's key center $c_k$. We use keys only for assignment because, in dot-product attention, retrieval weights depend on $q^\top k$ while values affect only the retrieved content after weighting (Vaswani et al., 2017); values are typically more content-variant and thus less reliable for stable online grouping. (ii) *Spatial continuity*: $\lambda_{\mathrm{sp}} d_{\mathrm{Mah}}(s_i; \mu_k, \Sigma_k)$ enforces track-consistent association under prototype $k$'s running location distribution $(\mu_k, \Sigma_k)$, acting as a prototype-specific elliptical gate that reduces spurious switching when multiple similar objects co-occur (Wojke et al., 2017). (iii) *Staleness control*: $\lambda_{\mathrm{idle}} \mathbf{1}[idle(\tau_k)]$ discourages matching to prototypes that have not been updated recently, where $idle(\tau_k)$ denotes the predicate $t - \tau_k > T_{idle}$ with $t$ being the current stream time. Once the bank is populated, continuity-aware assignment reduces prototype switching and fragmentation even when many updates occur after the decisive cue, preventing evidence from being scattered and diluted across prototypes.

**Prototype update.** After assigning token $i$ to $k^\star$, we update the selected entry by (i) updating centers $(c_{k^\star}, p_{k^\star})$ (e.g., via EMA), (ii) incrementing mass $n_{k^\star}$, and (iii) updating residual statistics $R_{k^\star}$ using the token's key/value residuals. We also update the spatial state $(\mu_{k^\star}, \Sigma_{k^\star})$ with $s_i$ and refresh recency metadata $\tau_{k^\star}$. This assignment is lightweight, online, and query-agnostic and is orthogonal to our query-time pseudo-token interface; alternative online association rules can be substituted without changing the pseudo-token mechanism (see Algorithm 1 for details).

**Capacity management.** Beyond per-token updates, ProtoKV applies lightweight maintenance to keep the bank stable over long streams (Algorithm 2). Idle prototypes whose recency $\tau_k$ has not been refreshed for $T_{idle}$ steps

*Table 1.* Main benchmark performance on SVU benchmarks. We use token budget $|M|$ = 24k tokens for RVS-Ego and RVS-Movie, and $|M|$ = 4k tokens for OVO-Bench and StreamingBench. For StreamingBench, we evaluate only Real-time visual understanding queries. Score columns are on a 0-5 scale.

| Backbone | Method | RVS-Ego | | RVS-Movie | | OVO-Bench | StreamingBench |
| --- | --- | --- | --- | --- | --- | --- | --- |
| | | Acc. (%) | Score | Acc. (%) | Score | Acc. (%) | Acc. (%) |
| LLaVA-OV-7B | SWA | 55.7 | 3.30 | 50.8 | 3.40 | 53.5 | 72.9 |
| | InfiniPot-V | 57.8 | 3.50 | 51.4 | 3.50 | 54.2 | 75.2 |
| | **ProtoKV** | **58.6** | **3.63** | **52.1** | **3.59** | **54.8** | **76.3** |
| Qwen2.5-VL-7B | SWA | 58.4 | 3.60 | 52.0 | 3.55 | 53.2 | 75.9 |
| | InfiniPot-V | 58.9 | 3.75 | 52.3 | 3.69 | 53.6 | 76.4 |
| | **ProtoKV** | **59.4** | **3.84** | **53.1** | **3.77** | **54.4** | **77.3** |

undergo mass decay, redundant prototypes whose centers are sufficiently close are merged, and slots whose mass has decayed to zero are recycled by reinitializing from a recent token in $B_near$. This preferentially preserves frequently supported prototypes while keeping the policy fixed across experiments. Since decisive cues are first retained exactly in the near window and then absorbed via continuity-aware assignment, prototypes that continue receiving consistent support quickly accumulate mass and are unlikely to be recycled immediately.

### 3.3. Residual Statistics and Pseudo-Token Interface

Prototype centers $(c_k, p_k)$ are a strong constant-memory summary, but center-only aggregation is fundamentally lossy: as more tokens are absorbed, distinctive cues are averaged out. This matters most under large query delay $\Delta$, where evidence must remain usable after many post-evidence updates: this is precisely the regime where decisive evidence becomes temporally distant and far-memory quality governs whether it remains usable.

**Residual statistics.** For a token $i$ assigned to prototype $k$, we define residuals: $r_i^K = K_i - c_k, \quad r_i^V = V_i - p_k$. The key point is not to reconstruct individual tokens, but to retain intra-prototype diversity that a single center cannot represent. We implement this with Product Quantization (Jegou et al., 2010)-based streaming histogram statistics: it offers a fixed-budget summary that (i) can be updated online without storing per-token codes, and (ii) captures multi-pattern residual structure around the center, reducing the chance that decisive cues are washed out as updates accumulate.

**Pseudo tokens.** Residual statistics are useful only if they can be consumed at query time without changing the model. We expose far memory as a fixed set of pseudo tokens: synthetic KV-cache entries that are directly consumable by standard attention. Injecting a small set of auxiliary key/value vectors as a lightweight interface to attention has been explored in parameter-efficient adaptation (Li & Liang, 2021). Unlike learned prefixes, our pseudo tokens are syn-

thesized online in a training-free manner from the prototype bank statistics. ProtoKV converts each prototype into $S$ pseudo tokens by combining its center with representative residual patterns implied by the stored statistics $R_k$. These pseudo-tokens are concatenated with the near-window exact KV and processed by standard attention unchanged, resulting in a fixed-length query context. This design keeps the per-query context size predictable, controlling query-time latency and attention cost even when decisive evidence lies in far history. For positional encoding, all $S$ pseudo-tokens synthesized from prototype $k$ share the same rotary position $\tau_k$, the original stream position of the most recently absorbed source token, while near-window tokens retain their original positions. This preserves prototype-level temporal recency without per-pseudo-token position tracking and keeps the interface fully compatible with standard attention (see Section D.1 for a comparison against alternative anchor choices).

**Mass-aware bias.** A prototype summarizes $n_k$ absorbed tokens; without accounting for multiplicity, far memory can underweight frequently observed evidence relative to rare prototypes. We therefore apply a simple per-prototype log-mass bias $b_k = \log n_k$ to the attention logits. For each attention head, the logit of pseudo-token $s$ decoded from prototype $k$ becomes

$$\ell_{k,s} = \frac{q^\top \tilde{k}_{k,s}}{\sqrt{d}} + b_k, \qquad (2)$$

where $b_k$ is shared across all $S$ pseudo-tokens of prototype $k$ and applied before the softmax. This approximates the effect of having $n_k$ duplicated tokens without instantiating them, so that a prototype's influence in attention is proportional to its accumulated evidence. This log-count additive correction is analogous to prior-based logit adjustments under softmax normalization (Menon et al., 2021; Ren et al., 2020).

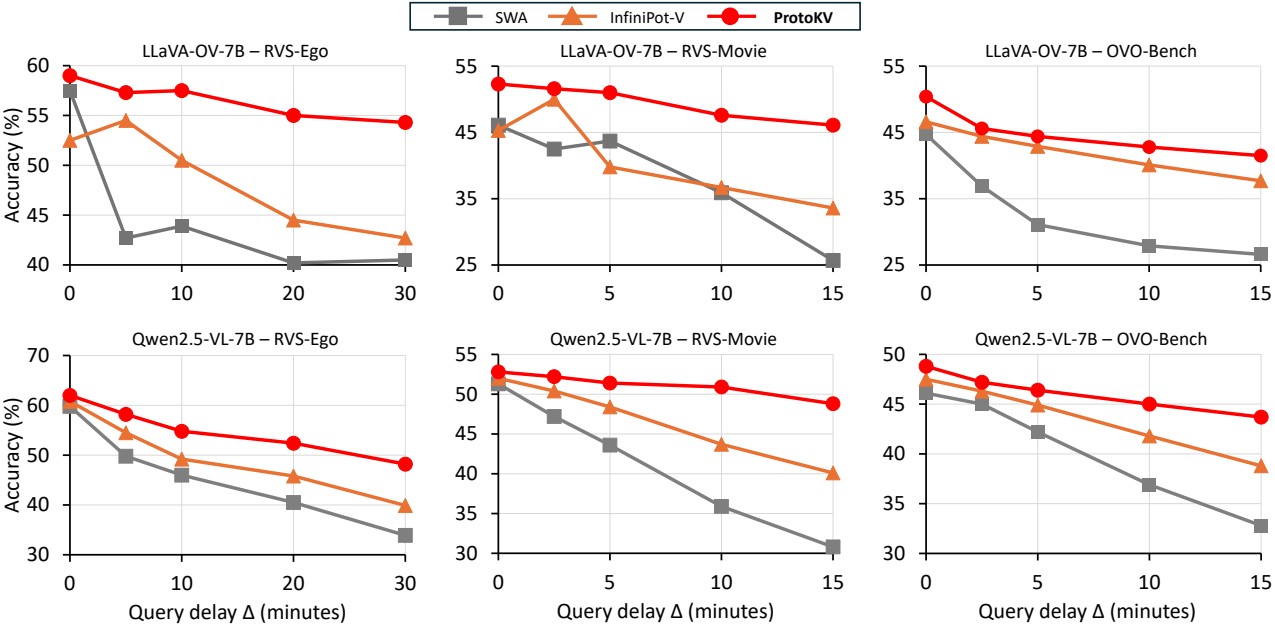

*Figure 3.* Query-delay robustness under matched token budgets in the online SVU setting. We apply a delayed-query sweep by setting $t_q = t_0 + \Delta$ (minutes) and retain only retrospective query types whose answers are invariant to shifting $t_q$. For fair comparisons across $\Delta$, we evaluate on the intersection subset that is valid for all delay values. We use token budget $|M|$ = 24k tokens for RVS-Ego and RVS-Movie, and $|M|$ = 4k tokens for OVO-Bench.

## 4. Experiments

### 4.1. Experimental Setup

**Benchmarks.** We evaluate on four SVU benchmarks (Zhang et al., 2025; Niu et al., 2025; Lin et al., 2024) that provide a dataset-defined query timestamp for each example and task-type annotations that allow principled grouping of queries. We follow the official evaluation split for each dataset. These annotations enable controlled delayed-query sweeps and rule-based filtering of query types without manual curation. For RVS-Ego and RVS-Movie (Zhang et al., 2025), we follow the benchmark's official evaluation pipeline and use gpt-3.5-turbo-0125 to compute the answer score on a 0–5 scale, where higher is better.

**Delayed-Query Protocol and Valid-Query Selection.** Ideally, one would evaluate query delay relative to the last-seen evidence time $t_e$ by setting $t_q = t_e + \Delta_e$. Since the evaluated SVU benchmarks do not provide $t_e$ per query, we instead use the dataset-defined query timestamp $t_0$ as a reference and sweep delayed queries by setting $t_q = t_0 + \Delta$.

To keep ground-truth unchanged under this shift, we apply a rule-based valid-query filter and evaluate only retrospective query types whose answers depend on past video content rather than on events occurring after $t_0$. For query types that are inherently time-dependent (e.g., future-facing or real-time queries), we do not include them in query-delay robustness experiments. See Appendix C.1 for further de-

tails.

For fair comparisons across delay values, we evaluate on the intersection subset of queries that remain valid for all $\Delta$ in the sweep. Concretely, for each example with video length $L$, we keep it only if $t_0 + \Delta \leq L$ holds for every evaluated $\Delta$, ensuring that all delay points are computed on the same set of queries. All $\Delta$ values reported in the experiments are measured in minutes.

**Baselines.** We restrict our comparisons to methods that match our deployment constraints: (i) bounded KV memory, (ii) training-free evaluation (no additional fine-tuning), and (iii) query-agnostic online updates, where the memory state is continuously maintained as frames arrive and queries are answered immediately at tq without query-conditioned retrieval or re-encoding. Under these criteria, we include two representative designs. First, Sliding-window attention (SWA) serves as the most naive bounded-memory baseline: it guarantees constant memory simply by retaining only the most recent window of KV pairs and truncating all earlier history. Second, query-agnostic continual KV token retention (Kim et al., 2025) maintains a fixed KV budget under a hard cap by selecting, merging, or evicting token instances online. We instantiate ProtoKV on two 7B-scale vision-language backbones, LLaVA-OV-7B (Li et al., 2025) and Qwen2.5-VL-7B (Bai et al., 2025) with identical pretrained weights and no fine-tuning unless noted.

**Budget Matching and Efficiency Measurement.** We com-

pare methods under a common query-time context budget by controlling the number of visual tokens that each method allows the model to attend to at query time. We then report the resulting measured peak GPU memory and latency metrics to verify that the compared settings fall within the intended budget regime. Query-time latency is reported separately in Fig. 4b.

## 4.2. Main Benchmark - SVU

We first evaluate ProtoKV on standard SVU benchmarks under matched peak GPU memory; query-time latency is validated separately in Fig. 4b under the same budgeted setting. Table 1 reports main benchmark accuracy and scores for two backbones and compares ProtoKV against SWA and a token-retention baseline.

ProtoKV achieves the best overall performance across the RVS benchmarks on both backbones, consistently outperforming SWA and the token-retention baseline. ProtoKV likewise yields higher accuracy on OVO-Bench and StreamingBench under the same KV cache memory budget. Overall, these results indicate that ProtoKV's summary-state far memory makes more effective use of limited memory than window truncation or token-instance retention, especially when decisive evidence must be preserved over long update horizons.

## 4.3. Query-Delay Robustness

Fig. 3 reports accuracy as a function of query delay $\Delta$. Across all settings, performance decreases as $\Delta$ grows, but the degradation profile differs sharply by memory mechanism. SWA exhibits an abrupt drop once the delay pushes the relevant evidence beyond the near window, reflecting that evidence is irrecoverably lost after truncation. InfiniPot-V degrades more gradually, yet still shows a consistent downward trend as post-evidence updates accumulate under a fixed budget. In contrast, ProtoKV remains substantially more stable over long delays: the slope of degradation is noticeably flatter, and the relative gains over both SWA and token retention increase with $\Delta$—precisely in the regime where SVU must rely on far memory rather than near-window access.

At the largest evaluated delays, across both backbones, ProtoKV yields up to +20.4pp over SWA and +12.5pp over token retention on RVS-Movie, with comparable long-delay improvements on RVS-Ego and OVO-Bench. This behavior supports our central interpretation of $\Delta$ as post-evidence update pressure, not merely "older evidence": larger $\Delta$ implies the decisive cue must survive more overwrite/eviction/merge events under the same capacity. These results indicate that ProtoKV's fixed-capacity summary-state updates mitigate brittleness under long delays while preserving bounded query-time cost.

*Table 2.* Component ablations under matched peak GPU memory ($|M| = 24$k, model: LLaVA-OV-7B, dataset: RVS-Ego). Evaluated on the same intersection subset used in Figure 3.

| Setting | Acc.@$\Delta=0$ (%) | Acc.@ $\Delta=30$ (%) |
|---|---|---|
| **ProtoKV (full)** | 59.0 | **54.3 (-4.7)** |
| *Memory tier* | | |
| Near only (w/o far) | 57.5 | 40.5 (-17.0) |
| Far only (prototype) | 54.5 | 49.2 (-5.3) |
| *Far-memory expression* | | |
| w/o residual stats | 58.5 | 51.0 (-7.5) |
| w/o mass bias ($b_k=0$) | 58.3 | 51.0 (-7.3) |
| *Assignment* | | |
| w/o continuity ($\lambda_{sp}=0$) | 56.2 | 47.5 (-8.7) |

*Table 3.* Performance on offline video understanding benchmark (OVU) under fixed memory budgets. * denotes the numbers from official paper. We use Qwen2-VL-7B in this table for alignment with prior OVU baselines.

| Compression Method | Budget $|M|$ | VideoMME Acc. (%) | MLVU Acc. (%) |
|---|---|---|---|
| DyCoke* | 3K | 55.3 | 57.5 |
| | 6K | 59.7 | 60.5 |
| InfiniPot-V* | 3K | 61.2 | 67.2 |
| | 6K | 62.8 | 68.4 |
| **ProtoKV** | 3K | **62.1** | 67.0 |
| | 6K | **63.7** | **68.9** |

## 4.4. Ablation Study

Table 2 summarizes component ablations under matched peak GPU memory. ProtoKV shows a relatively mild drop as query delay increases, while the ablations reveal that this robustness is not due to any single trick but to the overall design. In particular, removing the far-memory summary leads to a pronounced failure at long delays, whereas keeping only the far prototypes preserves delayed-query behavior but sacrifices performance in the near regime. This highlights that ProtoKV's robustness arises from the intended complementarity between an exact near window for immediate queries and a durable far summary for delayed queries.

Among far-memory components, removing residual statistics or mass-aware weighting worsens delayed-query performance, and disabling continuity-aware association further increases degradation, aligning with the need for both expressive summaries and stable online maintenance to avoid evidence fragmentation under sustained updates. Consistent with this, ProtoKV's long-delay stability depends jointly on (i) preventing over-averaging in the far summary and (ii) maintaining stable assignment over time under a hard budget.

## 4.5. Auxiliary Evaluation on Offline Video Understanding

To complement our SVU evaluation, we additionally evaluate ProtoKV on offline video understanding benchmarks (Fu et al., 2025; Zhou et al., 2025) under the same memory budget, using Qwen2-VL-7B (Wang et al., 2024) to align with prior OVU baselines' reported settings. We compare ProtoKV against representative OVU baselines including DyCoke (Tao et al., 2025). As shown in Table 3, ProtoKV improves performance on VideoMME under tight budgets and remains competitive on MLVU, staying close to the best baseline across budgets. Overall, these results suggest that ProtoKV can be effective beyond SVU-specific evaluations under memory-constrained offline video understanding.

## 4.6. Task-Type Analysis

To examine where summary-state far memory helps and where it falls short, we decompose ProtoKV's behavior along MLVU's task taxonomy (Section D.2). ProtoKV is most reliable on tasks whose answers depend on localized state cues or relations among visually distinct events. The clearest weakness appears in tasks that require separating or counting many visually similar repeated events: under a fixed prototype capacity, such occurrences are absorbed into the same prototype and their boundaries are blurred by compression—a mechanistic trade-off of representing far history as a fixed-capacity summary rather than as token instances. Representative cases are provided in Section E.

## 4.7. System Validation

Fig. 4 summarizes the system-level budget sensitivity of ProtoKV under memory constraints. Fig. 4a reports accuracy as we vary the memory budget, showing that ProtoKV consistently delivers the strongest accuracy among the budgeted baselines, with larger budgets translating into more reliable improvements. Fig. 4b compares query-time TTFT (time from query arrival to the first generated token) under a matched memory budget. ProtoKV attains competitive query-time TTFT relative to budgeted baselines, and is substantially faster than DyCoke (Tao et al., 2025), an OVU-oriented, query-conditioned reference method whose query-time selection introduces additional overhead. Overall, Fig. 4 validates that ProtoKV achieves favorable accuracy–latency trade-offs under fixed memory budgets, aligning with the low-latency requirements of streaming scenarios.

**Comparison to SVU offloading baseline.** Table 4 compares ProtoKV with ReKV (Di et al., 2025), a representative offloading/retrieval-based SVU baseline, to contextualize the latency–accuracy trade-off against an external-memory approach under matched peak GPU memory. While achiev-

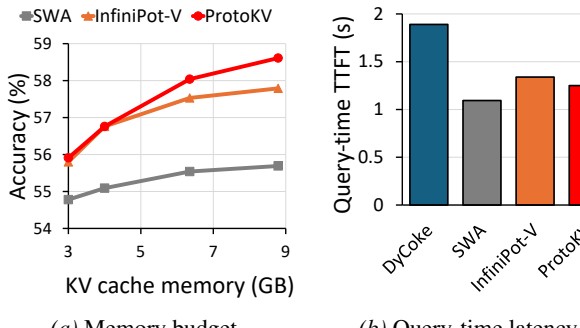

*(a)* Memory budget     *(b)* Query-time latency

*Figure 4.* Budget sensitivity and query-time TTFT under memory constraints. (a) Accuracy versus memory budget. (b) Query-time TTFT (query arrival → first token) under a matched memory budget. DyCoke is shown as a query-conditioned reference.

*Table 4.* Comparison to an SVU offloading/retrieval baseline on RVS-Ego under matched peak GPU memory with LLaVA-OV-7B. TTFT (query arrival → first token) and E2E latency (query arrival → completion) are measured.

| Compression Method | Acc. (%) | Peak GPU (GB) | TTFT (s) | E2E Lat. (s) |
|---|---|---|---|---|
| ReKV | 58.8 | 23.0 | 1.43 | 3.92 |
| **ProtoKV** | 58.0 | 23.0 | 1.06 | 2.90 |

ing comparable accuracy on RVS-Ego, ProtoKV improves query-time responsiveness, reducing both query-time TTFT and end-to-end latency by 26%. This indicates that a GPU-resident constant-footprint memory can better satisfy the low-latency requirements of streaming SVU with asynchronous queries and strict response-time budgets.

**Online update overhead.** Beyond query-time TTFT, we measure ProtoKV's online cost during stream ingestion on Qwen2.5-VL-7B (RTX 5090, RVS-Ego), and find a per-frame update overhead of 33.8 ms, about 30.8% of total per-frame processing time (Section C.3). This cost is incurred during ingestion rather than at query time and can be partially overlapped with inter-frame gaps in sampled-frame SVU pipelines, so it does not directly affect query-time TTFT.

## 5. Related Work

**SVU with bounded memory.** A common streaming baseline is sliding-window KV caching, which guarantees constant memory but can discard decisive cues when queries arrive after long and unpredictable delays. InfiniPot-V (Kim et al., 2025) extends this regime with online token-retention/compression under a fixed cap, mitigating outright eviction yet still representing far history as a subset of token instances — a limitation our summary-state design directly targets. As a complementary approach, ReKV (Di et al., 2025) attempts to mitigate information

loss by selectively recalling critical KV pairs, though its efficiency is often bounded by the requirement of a relatively large memory budget to sustain accuracy. Alternatively, ProVideLLM (Chatterjee et al., 2025) utilizes a multimodal interleaved cache to represent distant history through verbalized text tokens, though it necessitates specialized training for visual-text alignment. Parallel to cache-centric methods, VideoLLM-MoD (Wu et al., 2024) optimizes computational efficiency by dynamically skipping layer-wise processing for redundant visual tokens via a mixture-of-depths approach.

**Offline Video Understanding and Long-Context VLMs.**
Long-context VLMs typically compress video tokens in offline settings where the full sequence is accessible. LongVU (Shen et al., 2025) employs a training-based adaptive compression module, while DyCoke (Tao et al., 2025) offers a training-free alternative via importance-based pruning. However, these offline-centric designs are not optimized for the continuous update pressure of streaming SVU, where fixed memory budgets must sustain sparse, decisive evidence over indefinite horizons without full-sequence re-evaluation.

**KV Cache Compression in Large Language Models.** KV cache management in LLMs primarily relies on sliding windows (Xiao et al., 2024) or importance-based eviction (Li et al., 2024a; Zhang et al., 2023). While effective for discrete linguistic tokens, these token-selection strategies are often too brittle for video streams. As demonstrated in our experiments, such selection-only approaches fail to preserve sparse visual cues when subjected to long evidence delays and sustained post-evidence updates, a gap our summary-state approach directly addresses.

## 6. Conclusion

In this work, we address streaming video understanding under strict memory budgets, where decisive evidence can appear briefly but must remain usable after many subsequent updates before a query arrives. We propose ProtoKV, a constant-footprint KV memory that keeps an exact near-window cache while representing far history as an object-centric prototype bank with lightweight residual statistics. At query time, ProtoKV exposes this summary state through a bounded pseudo-token interface compatible with standard attention, keeping query-time computation controlled. Experiments on SVU benchmarks show that ProtoKV achieves competitive or improved accuracy under matched budgets and remains significantly more robust as query delay grows, with larger gains in the long-delay regime. These results highlight the value of summary-state far memory for stable streaming systems under sustained update pressure, and motivate future work on richer online summaries and broader SVU settings.

## Acknowledgement

We appreciate anonymous reviewers for providing constructive feedback and suggestions. This work was supported by Samsung Electronics, Institute for Information & communications Technology Planning & Evaluation (IITP) grant funded by the Korea government (MSIT)(RS-2019-II190075,Artificial Intelligence Graduate School Support Program (KAIST)), and National Research Foundation of Korea (NRF) grant funded by the Korea government (MSIT) (No.RS-2024-00340099).

## Impact Statement

This paper presents work whose goal is to advance the field of machine learning, specifically improving the memory and latency efficiency of streaming video understanding. By lowering deployment cost, this can broaden access to video-capable models in resource-constrained settings. There are many potential societal consequences of our work, none of which we feel must be specifically highlighted here.

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

# A. ProtoKV Method Details

## A.1. Streaming Memory State and Residual Representation

ProtoKV's bounded, query-agnostic streaming memory has a two-tier state: a near ring buffer $B_{\mathrm{near}}$ storing the most recent $W$ tokens' exact KV pairs, and a fixed-capacity far prototype bank

$$B_{\mathrm{far}} = \{\Pi_k\}_{k=1}^{K_{\max}}, \quad \Pi_k = (c_k, p_k, n_k, R_k, \mu_k, \Sigma_k, \tau_k),$$

where $(c_k, p_k)$ are key/value centers, $n_k$ is the prototype mass, $R_k$ stores residual statistics, $(\mu_k, \Sigma_k)$ are running spatial statistics, and $\tau_k$ records the stream position of the most recently absorbed source token. Each prototype thus summarizes many evicted tokens without retaining token instances.

**Residual statistics.** We implement $R_k$ as fixed-size Product Quantization (PQ) histograms over key and value residuals:

$$R_k = \left(H_k^K, H_k^V, n_k^{\mathrm{res}}\right), \quad H_k^K, H_k^V \in \mathbb{N}^{G \times C},$$

where $G$ is the number of PQ subquantizers and $C$ the codewords per subquantizer. Entry $H_k^K[g, c]$ counts how often a key residual assigned to prototype $k$ selects codeword $c$ in subspace $g$; $H_k^V$ is defined analogously for value residuals. The scalar $n_k^{\mathrm{res}}$ counts residual updates accumulated by prototype $k$. Default values are $G = 8$ and $C = 16$ (Section B.1).

**PQ codebook initialization.** The PQ codebooks $\{C_g^K, C_g^V\}_{g=1}^G$ are initialized via mini-batch $k$-means on a warm-up reservoir of residuals collected during the first $T_{\mathrm{warm}}$ streaming steps, run separately for each subspace and for key/value. Codebooks are shared across prototypes within the same layer/head and kept fixed thereafter.

## A.2. Streaming Update Procedure

As the stream progresses, each visual token is appended to $B_{\mathrm{near}}$; when the buffer overflows, the oldest token is evicted and absorbed into $B_{\mathrm{far}}$ via continuity-aware assignment (Section 3.2 of the main paper), and the selected prototype's centers, mass, recency, spatial statistics, and residual histograms are updated. Algorithm 1 gives the full procedure.

---

**Algorithm 1** ProtoKV Streaming Update

---

**Require:** Near buffer $B_{\mathrm{near}}$, far prototype bank $B_{\mathrm{far}} = \{\Pi_k\}_{k=1}^{K_{\max}}$, incoming visual token $(K_i, V_i, s_i)$ at time $t$
**Require:** Near-window budget $W$, PQ codebooks $\{C_g^K, C_g^V\}_{g=1}^G$
**Require:** Assignment weights $\lambda_{\mathrm{sp}}, \lambda_{\mathrm{idle}}$, idle threshold $T_{\mathrm{idle}}$, EMA rates $\alpha, \beta, \eta$
1: Append $(K_i, V_i, s_i, t)$ to $B_{\mathrm{near}}$
2: **if** $|B_{\mathrm{near}}| > W$ **then**
3:     Evict the oldest token $(K_e, V_e, s_e, t_e)$ from $B_{\mathrm{near}}$
4:     **if** there exists an inactive prototype slot $k$ **then**
5:         Initialize $\Pi_k$: $c_k \leftarrow K_e, p_k \leftarrow V_e, n_k \leftarrow 1, \mu_k \leftarrow s_e, \Sigma_k \leftarrow I, \tau_k \leftarrow t, R_k \leftarrow (\mathbf{0}^{G \times C}, \mathbf{0}^{G \times C}, 0)$
6:     **else**
7:         $k^\star \leftarrow \arg\min_k \left[ -\cos(K_e, c_k) + \lambda_{\mathrm{sp}} d_{\mathrm{Mah}}(s_e; \mu_k, \Sigma_k) + \lambda_{\mathrm{idle}} \mathbf{1}(t - \tau_k > T_{\mathrm{idle}}) \right]$
8:         $c_{k^\star} \leftarrow (1 - \alpha) c_{k^\star} + \alpha K_e$
9:         $p_{k^\star} \leftarrow (1 - \beta) p_{k^\star} + \beta V_e$
10:         $n_{k^\star} \leftarrow n_{k^\star} + 1, \tau_{k^\star} \leftarrow t$
11:         $\mu_{k^\star} \leftarrow (1 - \eta) \mu_{k^\star} + \eta s_e$
12:         $\Sigma_{k^\star} \leftarrow (1 - \eta) \Sigma_{k^\star} + \eta (s_e - \mu_{k^\star})(s_e - \mu_{k^\star})^\top$
13:         Update $R_{k^\star}$ from residuals $r^K = K_e - c_{k^\star}, r^V = V_e - p_{k^\star}$
14:     **end if**
15:     Apply PROTOTYPEMAINTENANCE($B_{\mathrm{far}}, t$)
16: **end if**
17:
18: **return** $(B_{\mathrm{near}}, B_{\mathrm{far}})$

---

After every absorption, ProtoKV applies three lightweight maintenance operations (Algorithm 2): *aging* decays the mass of idle prototypes; *merging* combines redundant prototypes whose key and value centers are sufficiently close; and *recycling* reinitializes prototype slots whose mass has decayed to zero, reseeding them from a recent token in $B_{\mathrm{near}}$.

---

**Algorithm 2** ProtoKV Prototype Maintenance

---

**Require:** Far prototype bank $B_{\text{far}} = \{\Pi_k\}_{k=1}^{K_{\max}}$, current time $t$, near buffer $B_{\text{near}}$
**Require:** Idle threshold $T_{\text{idle}}$, decay factor $\gamma$, merge thresholds $\varepsilon_K, \varepsilon_V$
1: **for** each prototype $\Pi_k \in B_{\text{far}}$ **do**
2:     **if** $t - \tau_k > T_{\text{idle}}$ **then**
3:         $n_k \leftarrow \lfloor (1-\gamma)n_k \rfloor$ {mass decay}
4:     **end if**
5: **end for**
6: **for** each pair $(\Pi_i, \Pi_j)$ with $i \neq j$ **do**
7:     **if** $\|c_i - c_j\|_2 < \varepsilon_K$ **and** $\|p_i - p_j\|_2 < \varepsilon_V$ **then**
8:         $c_i \leftarrow \frac{n_i c_i + n_j c_j}{n_i + n_j}, p_i \leftarrow \frac{n_i p_i + n_j p_j}{n_i + n_j}$
9:         $n_i \leftarrow n_i + n_j, H_i^K \leftarrow H_i^K + H_j^K, H_i^V \leftarrow H_i^V + H_j^V, n_i^{\text{res}} \leftarrow n_i^{\text{res}} + n_j^{\text{res}}$
10:         $\mu_i \leftarrow \frac{n_i \mu_i + n_j \mu_j}{n_i + n_j}$
11:         Mark $\Pi_j$ as inactive {merging}
12:     **end if**
13: **end for**
14: **for** each prototype $\Pi_k \in B_{\text{far}}$ **do**
15:     **if** $\Pi_k$ is empty (inactive or $n_k = 0$) **then**
16:         Select a recent token $(K_r, V_r, s_r)$ from $B_{\text{near}}$
17:         Reinitialize $\Pi_k$: $c_k \leftarrow K_r, p_k \leftarrow V_r, n_k \leftarrow 1, \mu_k \leftarrow s_r, \Sigma_k \leftarrow I, \tau_k \leftarrow t, R_k \leftarrow (\mathbf{0}^{G \times C}, \mathbf{0}^{G \times C}, 0)$ {recycling}
18:     **end if**
19: **end for**
20:
21: **return** $B_{\text{far}}$

---

## A.3. Query-Time Pseudo-Token Synthesis

At query time, ProtoKV exposes each prototype as $S$ pseudo-KV tokens decoded from $R_k$, concatenates them with the near-window KV, and feeds the result to standard attention. We treat each PQ histogram as a factorized categorical distribution over code tuples. The smoothed per-subspace probability for key residuals is

$$P_k^K(g, c) = \frac{H_k^K[g, c] + \epsilon}{\sum_{c'=1}^C H_k^K[g, c'] + C\epsilon}, \tag{3}$$

and a code tuple $z^K = (z_1^K, \ldots, z_G^K)$ scores $\sum_g \log P_k^K(g, z_g^K)$. The top-$S$ tuples are obtained by beam search over subquantizers (Algorithm 3), avoiding the $C^G$ exhaustive search. The decoded key residual for mode $s$ is $\bar{r}_{k,s}^K = \text{concat}_{g=1}^G C_{g, z_{k,s,g}^K}^K$; value residuals are decoded analogously from $H_k^V$. By default we decode key and value modes independently and pair them by rank, yielding pseudo-tokens

$$\tilde{k}_{k,s} = c_k + \bar{r}_{k,s}^K, \qquad \tilde{v}_{k,s} = p_k + \bar{r}_{k,s}^V.$$

This introduces a key–value independence approximation; a joint histogram $H_k^{KV} \in \mathbb{N}^{G \times C \times C}$ would decode paired modes directly at higher memory cost.

**Bounded attention context.** Concatenating pseudo-tokens from all $K_{\max}$ prototypes with the near-window KV yields a query-time context of fixed length $L_{\text{context}} = W + K_{\max} \cdot S$, independent of stream length. For each pseudo-token from prototype $k$, ProtoKV adds a log-mass bias $b_k = \log n_k$ to the attention logit before softmax:

$$\ell_{k,s} = \frac{q^\top \tilde{k}_{k,s}}{\sqrt{d}} + \log n_k, \tag{4}$$

shared across the $S$ pseudo-tokens of the same prototype. This approximates the multiplicity effect of summarizing $n_k$ source tokens, so a prototype's attention weight scales with accumulated evidence. Near-window tokens carry zero bias. Algorithm 4 ties these steps together.

---

**Algorithm 3** DecodeTopSResidualModes

---

**Require:** Residual histogram $H \in \mathbb{N}^{G \times C}$, PQ codebooks $\{C_g\}_{g=1}^G$, number of modes $S$, beam size $B$, smoothing $\epsilon$
**Ensure:** Top-$S$ decoded residual modes $\{\bar{r}_s\}_{s=1}^S$
 1: Initialize beam $\mathcal{B} \leftarrow \{(0, \emptyset)\}$ {log-score, partial code tuple}
 2: **for** $g = 1$ to $G$ **do**
 3:     $P(g, c) \leftarrow \dfrac{H[g, c] + \epsilon}{\sum_{c'=1}^C H[g, c'] + C\epsilon}$
 4:     $\mathcal{B}' \leftarrow \emptyset$
 5:     **for** each $(a, z_{1:g-1}) \in \mathcal{B}$ **do**
 6:         **for** $c = 1$ to $C$ **do**
 7:             $\mathcal{B}' \leftarrow \mathcal{B}' \cup \{(a + \log P(g, c), \ [z_{1:g-1}, c])\}$
 8:         **end for**
 9:     **end for**
10:     $\mathcal{B} \leftarrow \text{TopB}(\mathcal{B}')$
11: **end for**
12: $\{z_s\}_{s=1}^S \leftarrow \text{TopS}(\mathcal{B})$
13: **for** $s = 1$ to $S$ **do**
14:     $\bar{r}_s \leftarrow \text{concat}_{g=1}^G C_{g, z_{s,g}}$
15: **end for**
16:
17: **return** $\{\bar{r}_s\}_{s=1}^S$

---

**Algorithm 4** ProtoKV Query-Time Pseudo-Token Decoding and Attention

---

**Require:** Query matrix $Q$, near buffer $B_{\text{near}}$ with KV $(K_{\text{near}}, V_{\text{near}})$, far prototype bank $B_{\text{far}} = \{\Pi_k\}_{k=1}^{K_{\max}}$
**Require:** PQ codebooks $\{C_g^K, C_g^V\}_{g=1}^G$, pseudo-tokens per prototype $S$, beam size $B$, smoothing $\epsilon$
**Ensure:** Attention output $O$
 1: $\tilde{K} \leftarrow K_{\text{near}}, \tilde{V} \leftarrow V_{\text{near}}, b \leftarrow \mathbf{0}^{|K_{\text{near}}|}$
 2: **for** each active prototype $\Pi_k = (c_k, p_k, n_k, R_k, \ldots) \in B_{\text{far}}$ **do**
 3:     **if** $n_k^{\text{res}} = 0$ **then**
 4:         $\bar{r}_{k,s}^K, \bar{r}_{k,s}^V \leftarrow \mathbf{0}$ for all $s = 1, \ldots, S$
 5:     **else**
 6:         $\{\bar{r}_{k,s}^K\}_{s=1}^S \leftarrow \text{DECODETOPSRESIDUALMODES}(H_k^K, \{C_g^K\}_{g=1}^G, S, B, \epsilon)$
 7:         $\{\bar{r}_{k,s}^V\}_{s=1}^S \leftarrow \text{DECODETOPSRESIDUALMODES}(H_k^V, \{C_g^V\}_{g=1}^G, S, B, \epsilon)$
 8:     **end if**
 9:     **for** $s = 1$ to $S$ **do**
10:         $\tilde{k}_{k,s} \leftarrow c_k + \bar{r}_{k,s}^K, \quad \tilde{v}_{k,s} \leftarrow p_k + \bar{r}_{k,s}^V$
11:         $\tilde{K} \leftarrow [\tilde{K}; \tilde{k}_{k,s}], \quad \tilde{V} \leftarrow [\tilde{V}; \tilde{v}_{k,s}], \quad b \leftarrow [b; \ \log n_k]$
12:     **end for**
13: **end for**
14: $A \leftarrow Q\tilde{K}^\top / \sqrt{d} + b^\top$
15: $O \leftarrow \text{softmax}(A) \tilde{V}$
16:
17: **return** $O$

---

# B. Configuration and Hyperparameters

### B.1. Default Hyperparameter Configuration

Table 5 lists the default hyperparameter values used in all main experiments unless otherwise stated. The same defaults are applied across all six benchmarks in this paper, without per-dataset retuning. The only quantity varied across evaluation settings is the external memory budget $|M|$, which determines $W$ and $K_{\max}$ according to the scaling rule described below.

**Scaling rule for the memory budget.** When the external memory budget $|M|$ changes, the near-window size $W$ and the prototype capacity $K_{\max}$ are scaled together to preserve the default near–far ratio $W : (K_{\max} \cdot S) = 1 : 3$, with $S$ held fixed. All other hyperparameters in Table 5 remain unchanged across budget settings.

**Additional implementation constants.** A few additional constants appear in our implementation but are not critical to the main results: the warm-up window $T_{\mathrm{warm}}$ used to collect residuals for PQ codebook initialization (Section A) and the smoothing constant $\epsilon$ used in the PQ probability estimate (Equation (3)). Empirical performance is not sensitive to small perturbations of these constants within typical ranges; exact values are specified in our open-source release.

*Table 5.* Default hyperparameter configuration used in all main experiments. The same values are used across all six benchmarks; only $|M|$ varies across evaluation settings, which scales $W$ and $K_{\max}$ as described in the text.

| Group | Hyperparameter | Default |
|---|---|---|
| Memory state | Near : Far split, $W : (K_{\max} \cdot S)$ | 1 : 3 |
| | Pseudo-tokens per prototype, $S$ | 8 |
| Assignment | Spatial weight, $\lambda_{\mathrm{sp}}$ | 0.1 |
| | Idle weight, $\lambda_{\mathrm{idle}}$ | 0.01 |
| | Idle threshold, $T_{\mathrm{idle}}$ | 120 |
| Update (EMA) | Key/value center rate, $\alpha = \beta$ | 0.05 |
| | Spatial state rate, $\eta$ | 0.05 |
| Maintenance | Idle decay rate, $\gamma$ | 0.05 |
| | Key/value merge thresholds, $(\varepsilon_K, \varepsilon_V)$ | (0.20, 0.25) |
| Residual statistics | PQ subquantizers, $G$ | 8 |
| | Codewords per subquantizer, $C$ | 16 |
| | Beam size, $B$ | $4S \ (= 32)$ |

## B.2. Hyperparameter Sensitivity

ProtoKV exposes hyperparameters in three groups: (i) *budget allocation*, which divides the query-time token budget $|M| = W + K_{\max} \cdot S$ between the near window and the far prototype bank; (ii) *assignment and refresh* ($\lambda_{\mathrm{sp}}, \lambda_{\mathrm{idle}}, T_{\mathrm{idle}}$), which control prototype continuity and staleness behavior; and (iii) *update and maintenance*—the EMA rates ($\alpha, \beta$) and the idle-decay rate $\gamma$—which control long-run bank stability. We fix a single default configuration (Section B.1) for all main results and vary only the external token budget when reporting scaling trends, so reported differences reflect the memory mechanism rather than per-dataset retuning. This subsection evaluates ProtoKV's sensitivity around this default operating point.

**Budget allocation.** Under a fixed total budget $|M| = 24$k, we vary the near–far split by adjusting $W$ versus $K_{\max} \cdot S$ while holding $|M| = W + K_{\max} \cdot S$ constant. We also vary the per-prototype pseudo-token count $S$ with $K_{\max} \cdot S$ fixed. Results on Qwen2.5-VL-7B and RVS-Ego are reported in Table 6. Performance is similar across balanced splits, with only the extreme near-heavy setting (3:1) degrading at long delay because the far-memory budget becomes too small to retain delayed evidence. The pseudo-token count $S$ is more influential, as it directly controls the trade-off between the prototype count $K_{\max}$ and per-prototype representation granularity under the fixed budget; we find $S = 8$ to balance the two. Our default 1:3 split was chosen to match the InfiniPot-V comparison setting rather than tuned per benchmark.

**Update and maintenance coefficients.** We additionally perturb each key update and maintenance coefficient by $0.5\times$ and $2\times$ its default value on both RVS-Ego and RVS-Movie, keeping all other settings at default. Table 7 reports the resulting accuracy changes relative to the default. Across all swept coefficients, the changes remain modest, staying within 0.9 points of the default on both datasets. This indicates that ProtoKV is not highly sensitive to these coefficients around its default setting, supporting our use of the same defaults across all six benchmarks without per-domain retuning.

*Table 6.* Sensitivity to budget allocation under fixed $|M| = 24k$ on Qwen2.5-VL-7B / RVS-Ego. Accuracy is reported at query delays $\Delta = 0$ and $\Delta = 30$ minutes. Top: varying the near-window vs. far-memory split under a fixed total budget. Bottom: varying the pseudo-token count $S$ per prototype with $K_{\max} \cdot S$ held fixed.

| $W : (K_{\max} \cdot S)$ | Acc. @ $\Delta = 0$ | Acc. @ $\Delta = 30$ |
|---|---|---|
| 1:5 | 61.2 | 48.5 |
| 1:3 (default) | 61.0 | 48.2 |
| 1:2 | 60.6 | 48.0 |
| 3:1 | 61.0 | 43.2 |
| $S = 4$ | 57.7 | 46.0 |
| $S = 8$ (default) | 61.0 | 48.2 |
| $S = 16$ | 58.8 | 46.7 |

*Table 7.* Sensitivity to update and maintenance coefficients on RVS-Ego and RVS-Movie under the default budget and Qwen2.5-VL-7B. Entries are accuracy changes (points) relative to the default setting at $\Delta = 0$. Default accuracy at $\Delta = 0$ is 61.0 on RVS-Ego and 53.3 on RVS-Movie.

| | RVS-Ego | | RVS-Movie | |
|---|---|---|---|---|
| Param (default) | $\times 0.5$ | $\times 2$ | $\times 0.5$ | $\times 2$ |
| $\alpha, \beta$ (0.05, 0.05) | $-0.3$ | $-0.5$ | $-0.4$ | $-0.6$ |
| $\lambda_{\mathrm{sp}}$ (0.1) | $-0.4$ | $-0.8$ | $-0.5$ | $-0.4$ |
| $\lambda_{\mathrm{idle}}$ (0.01) | $-0.2$ | $-0.3$ | $-0.3$ | $-0.8$ |
| $\gamma$ (0.05) | $-0.7$ | $-0.8$ | $-0.2$ | $-0.5$ |
| $T_{\mathrm{idle}}$ (120) | $-0.2$ | $-0.9$ | $-0.4$ | $-0.4$ |

## C. Experimental Setting Details

### C.1. Dataset Selection for Query-Delay Sweep

**Delayed-query protocol and validity constraint.** Ideally, $t_0$ would be the last-seen timestamp of decisive evidence for each query; the SVU benchmarks evaluated here do not annotate this quantity, so we follow standard practice and use the dataset-provided query timestamp as a proxy for $t_0$. For a video of length $L$, a delayed query is valid only if $t_0 + \Delta \leq L$. To ensure fair comparisons across different $\Delta$ values, we evaluate on the intersection subset of queries that remain valid for all $\Delta$ in the sweep, so that every delay point is computed on the same set of queries.

**Retrospective query types per benchmark.** Shifting the query time from $t_0$ to $t_q$ can alter the ground truth for query types that depend on what is happening at the moment the query is posed. We therefore restrict the sweep to retrospective query types whose answers are determined by past video content and are invariant to the shift, using each benchmark's native task taxonomy:

- **RVS-Ego and RVS-Movie.** RVS (Zhang et al., 2025) categorizes queries into five answer types; we select the three types whose ground truth is determined by past events: *whether something happened*, *order judging*, and *what-event-order*.

- **OVO-Bench.** OVO-Bench (Niu et al., 2025) categorizes queries into three major types: forward active responding, real-time visual perception, and backward-tracing. We use only the *backward-tracing* subset, whose answers refer to past evidence that remains constant regardless of when the query is posed.

- **StreamingBench.** StreamingBench (Lin et al., 2024) is not included in the delay sweep. Its main benchmark setting in Table 1 restricts to real-time visual understanding queries (per the benchmark's own protocol), which by construction depend on the moment the query is posed and are not invariant to a query-time shift.

**Included vs. excluded examples.** Table 8 gives representative included and excluded queries for the benchmarks used in the delay sweep experiments.

*Table 8.* Examples of included and excluded queries for the query-delay sweep, paired by benchmark to highlight the contrast. Included queries are retained because their ground truth is determined by past video content and is invariant to a query-time shift; excluded queries reference the current moment and would change ground truth under the shift.

| Benchmark | Included (retrospective) | Excluded (current-moment) |
| --- | --- | --- |
| RVS-Ego | "Did the person wash the vegetables before turning on the stove?" *(Order Judging)* | "What is happening right now? Summarize the current scene." *(Scene Summary)* |
| OVO-Bench | "Earlier in the video, which object did the person pick up?" *(Backward-tracing)* | "What color is the object the person is holding now?" *(Real-time perception)* |

## C.2. Implementation Setup

This subsection documents two parts of the experimental setup that are needed to interpret the matched-budget comparisons: (i) the per-baseline allocation of the memory budget $|M|$, and (ii) the raw (pre-compression) visual context length under each backbone's streaming pipeline.

**Baseline configurations.**   All baselines are evaluated under the same benchmark protocol, video preprocessing pipeline, backbone, and total memory budget $|M|$; the memory mechanism is the only intended source of difference between methods.

- **SWA** (sliding-window attention) uses the entire memory budget as a single sliding window: given $|M|$, SWA retains the most recent $|M|$ tokens' exact KV pairs and truncates everything earlier. No additional hyperparameters are introduced.
- **InfiniPot-V** follows its original continual-compression setting. The total budget $|M|$ is split into a recent KV buffer of size $|M| - |C|$ and a far-memory cache of size $|C|$ with $|C|/|M| = 0.75$, where $|C|$ denotes the retained far-memory KV size. The update parameters $\alpha = 0.5$ and $\gamma = 0.125$ are used as in the original work; we do not retune these parameters for any of our evaluation settings.

ProtoKV's own configuration is reported separately in Section B.1 and is not repeated here.

**Raw sequence lengths.**   For a delayed query at shifted time $t_q = t_0 + \Delta$, the relevant pre-compression quantity is the raw visual prefix accumulated up to $t_q$. We report this quantity for each backbone so that the effective compression ratio against $|M|$ is explicit.

**LLaVA-OV setup.** We sample the video stream at 0.5 fps and produce 196 visual tokens per sampled frame, yielding a token arrival rate of roughly 98 tokens per second of video. Under this rate, the maximum raw visual prefix reached over the evaluated benchmarks is approximately 352,800 tokens on RVS-Ego, 198,200 tokens on RVS-Movie, and 205,600 tokens on OVO-Bench.

**Qwen2.5-VL setup.** We follow the same model-side sampling configuration as InfiniPot-V, which caps the raw visual context at 49,920 tokens before any compression. The Qwen2-VL-7B configuration used in Table 3 and Table 10 follows the same setup.

In both setups the raw context is then compressed to the budget $|M|$ by each bounded-memory method, so the effective compression factor scales with both stream length up to $t_q$ and the chosen $|M|$. The LLaVA-OV setting represents the more aggressive compression regime in our experiments, while the Qwen2.5-VL setting operates at a fixed raw ceiling regardless of stream length.

## C.3. Online Update Overhead

The main paper reports a per-frame online update cost of $33.8\,\mathrm{ms}$, about $30.8\%$ of the total per-frame processing time, measured on Qwen2.5-VL-7B with a single RTX 5090 GPU while streaming the RVS-Ego benchmark. This subsection provides the measurement setup and a per-stage breakdown that supports that number.

**Setup.**   We profile ProtoKV during stream ingestion (i.e., outside of any query). Each iteration of the streaming loop encodes one sampled frame into visual tokens, appends them to the near-window KV buffer, and applies ProtoKV's online update (eviction, continuity-aware assignment, prototype centers/mass/recency update, and residual histogram update) for

any tokens that fall outside the near window. Per-iteration latencies are averaged over a representative window of frames after a warm-up phase, with CUDA synchronization between stages to attribute time correctly. All measurements are taken under the default configuration listed in Section B.1.

**Where the update time goes.** Within the 33.8 ms per-frame update, the two largest contributors are the continuity-aware assignment and the prototype update (centers, mass, recency, and residual-histogram updates), each accounting for a substantial fraction of the total update time on this setup. The remaining maintenance routine is comparatively inexpensive because the prototype bank is small ($K_{\max}$ on the order of a few thousand under the default budget) and most maintenance steps reduce to constant-time checks. We do not separately profile the upstream visual encoder and KV append stages here; they are unchanged from the backbone's default streaming pipeline.

**Why this does not affect query-time TTFT.** The 33.8 ms update cost is incurred during stream ingestion rather than at query time, and the streaming loop runs continuously regardless of whether a query is pending. In sampled-frame SVU pipelines (e.g., 0.5 fps for LLaVA-OV in our setup, Section C.2), the inter-frame interval is on the order of seconds, which is much larger than the per-frame update cost. ProtoKV's update can therefore be overlapped with this inter-frame interval, and the work that contributes to query-time TTFT is just the bounded query-time attention over $W + K_{\max} \cdot S$ tokens, not the update cost. This separation is what allows ProtoKV to maintain low query-time TTFT even under continuous stream ingestion.

## D. Additional Experiments

### D.1. RoPE Position Anchor Comparison

The main paper states that each prototype $k$ shares a single rotary position $\tau_k$ across its $S$ pseudo-tokens, defined as the original stream position of the most recently absorbed source token assigned to that prototype. This subsection compares this default choice against two natural alternatives that aggregate the positions of the source tokens that have been absorbed into prototype $k$ over time:

- **First**: $\tau_k$ is set to the stream position of the *first* source token absorbed into the prototype, i.e., the position at which the prototype was initialized.

- **Average**: $\tau_k$ is set to the running mean of the stream positions of all source tokens absorbed into the prototype.

- **Most recent (default)**: $\tau_k$ is refreshed to the stream position of the most recently absorbed source token, as described in the main paper.

In all three variants, the $S$ pseudo-tokens of prototype $k$ share the same $\tau_k$, and near-window tokens retain their original positions; only the choice of which absorbed position $\tau_k$ tracks changes.

Table 9 reports accuracy under the three variants on Qwen2.5-VL-7B / RVS-Ego under the default memory budget. The recency-based anchor outperforms both alternatives, and the gap widens as the anchor reaches further into the past. The first-token anchor performs worst, consistent with the intuition that the position at which a prototype was *initialized* becomes progressively stale as the prototype continues to absorb new tokens, so by query time it no longer reflects when the supporting evidence was most strongly present. The average anchor partially mitigates this by drifting forward as new tokens are absorbed, but it still under-weights the most recent supporting evidence. The most-recent choice keeps $\tau_k$ close to the latest supporting evidence for each prototype, which is the temporal context in which downstream attention is most likely to use it.

*Table 9.* Accuracy under three RoPE position-anchor choices for $\tau_k$ on Qwen2.5-VL-7B / RVS-Ego under the default memory budget. All three variants share the same per-prototype anchor across the $S$ pseudo-tokens; they differ only in which absorbed position the anchor tracks.

| $\tau_k$ choice | Accuracy (%) |
|---|---|
| First absorbed position | 57.2 |
| Average absorbed position | 59.6 |
| Most recent (default) | **61.0** |

### D.2. MLVU Task-Type Breakdown

Table 10 reports ProtoKV's per-task accuracy on the seven MLVU task types under matched memory budget ($|M| = 6k$) and Qwen2-VL-7B. The pattern is consistent with the design of ProtoKV's summary-state far memory: prototypes preserve evidence about localized state and visually distinct events well, but they absorb repeated visually similar events into a single slot, which limits absolute accuracy on counting-style tasks.

*Table 10.* ProtoKV's per-task accuracy on MLVU under matched memory budget ($|M| = 6k$) and Qwen2-VL-7B. Task abbreviations: Order = Action Order, Count = Action Count, AR = Anomaly Recognition, Needle = Needle QA, Ego = Ego Reasoning, TR = Topic Reasoning. "Avg." is the official MLVU M-Avg score across the seven tasks.

| Task | Order | Count | AR | Needle | PlotQA | Ego | TR | Avg. |
|---|---|---|---|---|---|---|---|---|
| Acc. (%) | 54.8 | 36.9 | 68.5 | 79.7 | 74.4 | 65.9 | 86.3 | 68.9 |

### D.3. Dynamic Pseudo-Token Allocation

In the main paper we fix the number of pseudo-tokens per prototype to a constant $S$ across all prototypes, which gives every prototype the same query-time representational capacity and keeps the analysis uniform across the prototype bank. A natural question is whether unequal per-prototype allocation under the *same* total budget can improve performance: some prototypes summarize relatively static content with low residual variance, while others capture more dynamic content with higher variance, and the latter may benefit from finer exposure.

To probe this, we test a simple variance-aware allocation that keeps the total pseudo-token count fixed at $S_{\text{tot}} = K_{\max} \cdot S$ (so the query-time context length $L_{\text{context}}$ is unchanged) and assigns each prototype $k$ a share proportional to its residual variance:

$$S_k = S_{\text{tot}} \cdot \frac{\text{Var}(H_k)}{\sum_j \text{Var}(H_j)}.$$

On Qwen2.5-VL-7B and RVS-Ego under the default budget, variance-aware allocation improves accuracy over the fixed-$S$ baseline (Table 11). The gain is larger at long delay, consistent with the intuition that prototypes whose residual distributions concentrate around a few dominant patterns benefit from having more pseudo-tokens dedicated to those patterns when the query arrives long after the supporting evidence.

*Table 11.* Variance-aware pseudo-token allocation versus the fixed default ($S = 8$ per prototype) under matched total budget $S_{\text{tot}} = K_{\max} \cdot S$ on Qwen2.5-VL-7B / RVS-Ego.

| Setting | Acc. @ $\Delta = 0$ | Acc. @ $\Delta = 30$ |
|---|---|---|
| Fixed, $S = 8$ (default) | 61.0 | 48.2 |
| Variance-aware, $S_k$ adaptive | 61.7 | 50.0 |

The main paper retains the fixed-$S$ design because it instantiates ProtoKV's summary-state interface with the smallest set of moving parts: a uniform per-prototype representation, no extra allocation policy, and a query-time context length that is determined purely by the static configuration $(W, K_{\max}, S)$. The variance-aware variant above is therefore best read as evidence that the allocation budget itself is a useful additional degree of freedom for summary-state SVU memory, rather than as a competing design that the main results overlook. We leave a more thorough study of allocation policies—including stability across streams, interaction with prototype maintenance, and behavior under tighter overall budgets—to future work.

## E. Qualitative Case Studies

**Case 1 (success, Needle QA): localized state retrieval.**

> **Question.** What is the state of movement of the American toad at the mouth of the den in the video?
> **Ground truth.** Very little movement.
> **ProtoKV.** Very little movement.

The relevant cue is spatially localized to a small region in the frame and is consistently supported over a contiguous interval, so it can remain represented in the far-memory summary without needing to be disambiguated from many similar episodes.

A small number of prototypes carry enough mass and residual structure to preserve a localized state across sustained post-evidence updates.

**Case 2 (success, Plot QA): narrative continuation across distinct events.**

> **Question.** At the beginning of the video, a woman in a red coat and a man with a hat are talking in the car. What does the man do after he opens the car door and leaves?
> **Ground truth.** Make a phone call.
> **ProtoKV.** Make a phone call.

The question requires linking an early scene (a conversation in the car) to a later, visually distinct event (the man's action after leaving the car). Because the early conversation and the later action occur in different visual contexts, they tend to be absorbed into different prototypes under continuity-aware assignment, and the narrative thread connecting them can be reconstructed at query time from the surviving prototype centers and their relative recency $\tau_k$. ProtoKV identifies the correct continuation despite the intervening updates between the two scenes.

**Case 3 (failure, Action Order): order errors among visually similar events.**

> **Question.** Identify the option that corresponds to the order of events as they occur in the video.
> **Ground truth.** javelin throw $\rightarrow$ water sliding $\rightarrow$ abseiling $\rightarrow$ making jewelry.
> **ProtoKV.** water sliding $\rightarrow$ javelin throw $\rightarrow$ abseiling $\rightarrow$ making jewelry.

ProtoKV preserves the positions of *abseiling* and *making jewelry* but swaps the first two events. The two swapped events are both fast-motion outdoor sports and are visually more similar to each other than to the remaining events; under continuity-aware assignment, such events are more likely to share a prototype, in which case the prototype's recency anchor $\tau_k$ reflects only the most recently absorbed source token. The relative order between events absorbed into different prototypes is preserved through their $\tau_k$ values, but the relative order between events absorbed into the same prototype is not. The same case also shows that when the events *are* visually distinguishable (*abseiling*, *making jewelry*), ProtoKV preserves their positions correctly, consistent with Case 2.

**Case 4 (failure, Action Count): repeated visually similar events.**

> **Question.** In this video, how many instances are there of the "carving pumpkin" action scene in total?
> **Ground truth.** 5.
> **ProtoKV.** 1.

The five carving instances are visually similar to each other, so under continuity-aware assignment they are absorbed into the same prototype rather than into distinct slots. The prototype's mass $n_k$ grows with each absorbed instance and reflects accumulated evidence, but the boundaries between distinct episodes are not retained as separate slots. As a result, the model has access to strong evidence that the action occurred, but not to an explicit count of how many distinct episodes it occurred in.

