# OpenReview forum: "ProtoKV: Streaming Video Understanding under Delayed Query with Summary-State Memory"
_ICML.cc/2026/Conference — ICML 2026 regular_

### Official Review · Reviewer_5VsY · 2026-03-09

**Soundness:** 3
**Presentation:** 3
**Significance:** 3
**Originality:** 3
**Overall Recommendation:** 4
**Confidence:** 3

**Summary:**

This work studies streaming video understanding under strict memory budgets and proposes ProtoKV, which combines a near-window cache with a far-history memory. Specifically, the method represents distant history using an object-centric prototype bank augmented with lightweight residual statistics. Experiments show that ProtoKV outperforms prior baselines, especially when the delay between the relevant evidence and the query becomes large. The method also demonstrates reasonable generalization to offline video understanding benchmarks.

**Compliance With Llm Reviewing Policy:**

Affirmed.

**Final Justification:**

The rebuttal has addressed all of my concerns. I think this paper is acceptable, provided that all of these analyses and experiments in rebuttal are included in the final version.

**Key Questions For Authors:**

Please see the first and second questions in the weakness part. If the author can address my concern, I will raise my score.

**Limitations:**

yes

**Strengths And Weaknesses:**

**Strengths**
1. Training-free design. ProtoKV is a plug-and-play approach that does not require model fine-tuning or architectural changes. By synthesizing pseudo-tokens online, it can be integrated into existing vision-language models such as LLaVA and Qwen-VL.
2. Robustness to delayed evidence. Compared with prior approaches, ProtoKV shows stronger performance when the temporal gap between the relevant evidence and the query increases, which is an important property for streaming video understanding.
3. Generalization to offline video understanding. In addition to streaming settings, the method also achieves competitive results on offline video understanding benchmarks such as VideoMME and MLVU, suggesting a certain degree of generality.

**Weaknesses**
1. Sensitivity to hyperparameters. ProtoKV relies on a fairly large number of design choices and hyperparameters, including the near-window size $\(W\)$, prototype capacity $\(K_{\max}\)$, and several update / maintenance coefficients such as $\lambda_{sp}$, $\lambda_{idle}$, $T_{idle}$, $\alpha$ and $\beta$. However, the paper provides limited ablation or robustness analysis on how sensitive the method is to these choices, making it difficult to judge how stable the approach is across datasets and settings.
2. Potential loss of fine-grained long-term information. The far-memory design seems better suited for coarse event recall than fine-grained temporal retention. Since distant history is compressed into a fixed set of prototypes, subtle details may be lost, making it harder to answer queries about precise order, repeated occurrences, or the state at a particular moment. More discussions on this aspect should be included.
3. Limited success/failure analysis. The paper would be stronger with more detailed qualitative analysis of when ProtoKV succeeds or fails, especially on cases involving long-range temporal reasoning, repeated events, or subtle state changes.

---

> ### Author Rebuttal · Authors · 2026-03-29
>
> **W1. Hyperparameter sensitivity**
>
> We agree that hyperparameter sensitivity is important. ProtoKV does not rely on per-benchmark retuning: we use the same non-budget default hyperparameters across all six benchmarks in the paper, varying only the memory budget $|M|$ to match each evaluation setting. To further assess robustness around this default operating point, we test two groups of parameters: (i) budget-allocation parameters and (ii) other key update/maintenance coefficients.
>
> *Budget allocation.*
>
> Under a fixed $|M|=24$k, we test two budget-allocation variants on Qwen2.5-VL-7B, RVS-Ego: (i) varying the near--far split by adjusting $W$ versus $K_{\max}\cdot S$ while keeping $|M|=W+K_{\max}\ \cdot S$ constant, with $S=8$ fixed; and (ii) varying the pseudo-token count per prototype $S$ while keeping $K_{\max}\cdot S$ fixed.
>
> |W:(K_max*S)|Δ=0min.|Δ=30min.|
> |-|-|-|
> |1:2|61.2\%|48.5\%|
> |1:3 (default)|61.0\%|48.2\%|
> |1:5|60.6\%|48.0\%|
> |3:1|61.0\%|43.2\%|
>
> |S| Δ=0min.|Δ=30min.|
> |-|-:|-:|
> |4| 57.7\% | 46.0\% |
> |8 (default) | 61.0\% | 48.2\% |
> |16| 58.8\% | 46.7\% |
>
> Performance remains similar across the tested near--far splits, except that the extreme near-heavy setting degrades at long delay because the far-memory budget becomes too small. $S$ is somewhat more influential, as it directly controls the fixed-budget trade-off between prototype count and query-time representation granularity. Our default 1:3 split was chosen to match the InfiniPot-V comparison setting rather than tuned separately per benchmark.
>
>
> *Other key parameters*
>
> We also vary the main update/maintenance coefficients by 0.5× and 2× of their defaults on two datasets at $\Delta=0$:
>
> |Param (default)|Ego 0.5×|Ego 2×|Movie 0.5×|Movie 2×|
> |-|-|-|-|-|
> |α,β (0.05, 0.05)|-0.3|-0.5|-0.4|-0.6|
> |λsp (0.1)|-0.4|-0.8|-0.5|-0.4|
> |λidle (0.01)|-0.2|-0.3|-0.3|-0.8|
> |γ (0.05)|-0.7|-0.8|-0.2|-0.5|
> |T_idle (120)|-0.2|-0.9|-0.4|-0.4|
>
> The default accuracies are 61.0\% on RVS-Ego and 53.3\% on RVS-Movie.
>
> Across these sweeps, the changes remain modest, staying within 0.9 points of the default, suggesting that ProtoKV is not highly sensitive to these coefficients around its default setting.
>
> ---
> **W2. Fine-grained Long-range Information Loss.**
>
> Fine-grained long-range temporal retention is inherently challenging under bounded memory, since distant history must be compressed under a fixed budget. ProtoKV mitigates this trade-off through an exact near window, multiple far-memory prototypes, and prototype-level recency information.
>
>
> To directly probe this concern, we evaluate the corresponding MLVU task types—Action Order (precise order), Action Count (repeated occurrences), and Needle QA (state at a particular moment)—using only methods rerun in our own setup under a 6k budget (Qwen2-VL-7B).
>
> |MLVU task|Action Order|Action Count|Needle QA|
> |-|-|-|-|
> |ProtoKV|54.8\%|36.8\%|79.7\%|
>
> These results suggest that ProtoKV is not restricted to coarse event recall, though fine-grained retention remains more challenging for some evidence types than others. In particular, repeated-event counting remains difficult under bounded-memory compression, whereas precise order and moment-specific state can still be retained to a meaningful extent.
>
> ---
> **W3. Qualitative Analysis.**
>
> Our qualitative examples suggest that ProtoKV’s compressed far memory preserves localized state cues and relations among a small number of distinct events more reliably than many visually similar repeated episodes.
>
> *Success 1 (Needle QA / subtle state retrieval)*
>
> Q: “What is the state of movement of the American toad at the mouth of the den in the video?”
>
> GT/ProtoKV: Very little movement.
>
> This succeeds because the relevant cue is spatially localized and consistently supported over time, so it can remain represented in the far-memory summary without needing to separate many similar episodes.
>
> *Success 2 (Action Order / long-range temporal relation)*
>
> Q: “Please identify the option that corresponds to the order of events as they occur in the video”
>
> GT/ProtoKV: playing trombone → jetskiing → water sliding → abseiling.
>
> This succeeds because the question depends on temporal ordering across a few visually distinct events, which are less likely to be scattered or over-averaged than many repeated similar episodes.
>
> *Failure (Action Count / repeated events)*
>
> Q: “How many instances of the ‘carving pumpkin’ action appear?”
>
> GT: 5; ProtoKV: 1.
>
> Repeated visually similar actions can be absorbed into the same prototype, so the far-memory summary reflects accumulated evidence rather than the number of distinct episodes. As a result, prototype mass reflects accumulated evidence rather than an explicit count of repeated-event boundaries.
>
> ProtoKV can preserve subtle localized states and long-range relations among distinct events, but repeated-event boundaries can be blurred by compression.
>
> ---
> We thank the reviewer and will incorporate these clarifications in the revision.

---

> > ### Author Rebuttal · Reviewer_5VsY · 2026-04-01
> >
> > I thank the authors for their effort in the rebuttal. It has addressed all my concerns.

---

> > > ### Author Response · Authors · 2026-04-06
> > >
> > > We sincerely thank the reviewer for the thoughtful feedback and for taking the time to consider our rebuttal. We are glad that our response addressed the concerns.

---

### Official Review · Reviewer_xBx9 · 2026-03-12

**Soundness:** 3
**Presentation:** 3
**Significance:** 3
**Originality:** 3
**Overall Recommendation:** 4
**Confidence:** 4

**Summary:**

This paper addresses the problem of Delayed Evidence in Streaming Video Understanding. The authors propose ProtoKV, which abandons the traditional strategy of retaining specific token instances. Instead, it preserves precise KV caches within a recent sliding window, while compressing long-range history into a fixed-capacity Semantic-Spatial Prototype Bank, along with residual statistical information. During querying, these prototypes are decoded into a fixed number of pseudo-tokens, combined with logit biases that reflect their absorption counts, and seamlessly integrated into the standard attention mechanism of large models for inference. Overall, I believe this paper is interesting, but I would like to further discuss with the authors before determining my final score.

**Compliance With Llm Reviewing Policy:**

Affirmed.

**Final Justification:**

I appreciate the authors for solving my concerns, I will retain my positive score.

**Key Questions For Authors:**

1. Considering that some prototypes may absorb mostly static background information with low internal residual variance, while others capture complex dynamic actions with higher variance, have the authors considered dynamically allocating the number of pseudo-tokens across different prototypes based on their internal residual variance, in order to improve memory utilization efficiency?
2. ProtoKV performs assignment directly in the Key feature space of the pretrained VLM. If the base model itself has poor disentanglement in representing certain objects, would ProtoKV inherit and potentially amplify such mis-clustering errors?

**Limitations:**

Please see above.

**Strengths And Weaknesses:**

**Strengths**:
1. The overall design is fairly solid. Regardless of how long the video stream lasts, ProtoKV’s long-term memory remains bounded by a fixed number of prototypes, which offers clear advantages in both inference time and memory consumption.
2. When assigning old tokens to prototypes, the method incorporates a 2D spatial Mahalanobis distance. This tracking-like mechanism helps prevent features of the same object across consecutive frames from being fragmented and assigned to different prototypes.
3. The method demonstrates superior performance. As the temporal gap between key evidence and the query increases, SWA or token pruning methods exhibit a sharp performance drop, whereas ProtoKV degrades much more gracefully.

**Weaknesses**:
1. To maintain the stability of the prototype bank, the algorithm introduces numerous hand-crafted hyperparameters, including the recent window size, prototype capacity, number of pseudo-tokens, EMA update rate, etc. When applied to video domains with drastically different temporal dynamics—such as very fast or very slow scene transitions—the cost of tuning and generalizing these parameters may be high, potentially limiting adaptability to rapidly changing requirements.
2. To isolate and evaluate the impact of “delay” under controlled conditions, the authors explicitly filter out real-time queries such as “what is happening now” in their experiments. However, this separation may obscure how the mechanism generalizes to realistic mixed-query settings, where both real-time and delayed evidence questions coexist.

---

> ### Author Rebuttal · Authors · 2026-03-29
>
> **W1. Hyperparameter Tuning Cost across Domains.**
>
> In our experiments, ProtoKV did not require per-domain retuning across the six benchmarks in the paper. We keep the same non-budget default settings for the prototype update and maintenance parameters—including the EMA update rate and related coefficients—as well as the same near–far memory ratio (1:3) across the six benchmarks, despite their differing video domains and temporal dynamics, and vary only the external memory budget by evaluation setting.
>
> To assess robustness around this default operating point, we also ran sensitivity sweeps under a fixed budget. Performance remained similar across balanced near–far splits, with only the extreme near-heavy setting degrading at long delay because the far-memory budget became too small. Varying the main update/maintenance coefficients by 0.5× and 2× produced only modest changes around the default, while the pseudo-token count $S$ was somewhat more influential, as it directly controls the fixed-budget trade-off between prototype count and query-time representation granularity (see our response to Reviewer 5VsY, W1 for details).
>
> Together, these results indicate that ProtoKV does not incur costly per-domain hyperparameter tuning across the six tested benchmarks, and that the method remains reasonably stable under practical variations of its main parameters.
>
>
> ---
> **W2. Mixed-query Environment.**
>
> Our main benchmark evaluation (Table 1) already uses each benchmark’s original query set, without retrospective filtering. Retrospective filtering is applied only in the delay-sweep experiment (Fig. 3), whose purpose is to isolate the effect of delay under a controlled protocol; otherwise, shifting $t_q$ would change the ground truth for real-time queries. ProtoKV also performs well on these original benchmark settings (Table 1), indicating that the method is not limited to the filtered delay-sweep protocol.
>
> ---
> **Q1. Dynamic Pseudo-token Allocation.**
>
> We tested the reviewer’s suggestion with a variance-based dynamic allocation under the same query-time budget. Specifically, for each prototype $k$, we allocate pseudo-tokens in proportion to the variance of its residual statistics $H_k$,
> $\tilde S_k = S_{tot} \cdot Var(H_k) / \sum_j Var(H_j)$ with the total budget fixed at $S_{tot} = K_{max}\cdot S_{default}$ (Qwen2.5-VL-7B, RVS-Ego).
>
> | Setting | $\Delta$=0min. | $\Delta$=30min. |
> |-|-|-|
> | Fixed ($S_{default}=8$) | 61.0\% | 48.2\% |
> | Dynamic $S_k$ (variance-based) | 61.7\% | 50.0\% |
>
> Under the same total budget, the dynamic variant consistently outperforms the fixed baseline, with a larger gain at long delay. This supports the reviewer’s intuition that different prototypes may benefit from different pseudo-token budgets, especially when the answer depends more heavily on far memory. A plausible explanation is that prototypes with richer internal residual structure benefit from finer pseudo-token exposure, while simpler ones need fewer tokens. We therefore view dynamic pseudo-token allocation as a promising extension of ProtoKV.
>
> ---
> **Q2. Key Space Disentanglement.**
>
> ProtoKV performs assignment in the frozen Key space, so it can inherit mis-clustering from the pretrained backbone. However, it does not rely on key-only matching: the assignment rule also includes a spatial continuity term based on each prototype’s running spatial distribution ($\mu_k, \Sigma_k$), which helps separate co-occurring objects with similar Keys but different locations.
>
> This mitigation is supported by our ablation: in Table 2, removing the spatial term ($\lambda_{sp} = 0$) drops long-delay accuracy on RVS-Ego from 54.3\% to 47.5\%. Our claim is therefore narrower: ProtoKV does not eliminate backbone-level disentanglement errors, but the added spatial constraint reduces the chance that key-space ambiguity turns into persistent prototype mixing under sustained updates.
>
> ---
> We thank the reviewer for this constructive feedback and will reflect these points in the revision.

---

> > ### Author Rebuttal · Reviewer_xBx9 · 2026-04-01
> >
> > I appreciate the authors for solving my concerns, I will retain my positive score.

---

> > > ### Author Response · Authors · 2026-04-06
> > >
> > > Thank you for your careful reading of our rebuttal and for the positive feedback. We are glad that our response addressed your concerns.

---

### Official Review · Reviewer_R6tC · 2026-03-26

**Soundness:** 2
**Presentation:** 3
**Significance:** 2
**Originality:** 3
**Overall Recommendation:** 4
**Confidence:** 4

**Summary:**

This paper proposes ProtoKV, a training-free KV cache memory mechanism for streaming video understanding (SVU) designed to operate under strict constant-memory budgets while preserving critical, delayed visual evidence. Diverging from traditional sliding-window or instance-based token retention strategies, the authors introduce a two-tier memory architecture. Recent frames are kept in an exact near-window KV cache, while older history is continuously aggregated into a fixed-capacity summary state (prototype bank) with residual statistics via a continuity-aware assignment rule. During query time, these prototypes are synthesized into a bounded set of pseudo-tokens with mass-aware logit biases, making them drop-in compatible with standard attention mechanisms without requiring architectural changes. Empirical evaluations on SVU benchmarks demonstrate that ProtoKV effectively maintains long-term information, showing improved accuracy and robustness to long query delays compared to existing bounded-memory baselines, all while maintaining a constant query-time computational footprint.

**Compliance With Llm Reviewing Policy:**

Affirmed.

**Final Justification:**

the rebuttal has resolved my issues, so I raise my score to positive score, i.e., weak accept.

**Key Questions For Authors:**

1. To what extent do hyperparameter settings impact model performance, and how can the mechanism handle the broad distribution of in-the-wild scenarios?

2. Do the hand-crafted heuristic rules exhibit poor adaptability in certain complex streaming video understanding scenarios?

3. Flaws in Experimental Comparisons: There are notable flaws and incomplete details in the experimental comparisons.

4. How are positional encodings explicitly handled?

**Limitations:**

In the Impact Statement section, the authors mention that while this work improves long-horizon video comprehension, it may amplify privacy and surveillance risks. However, as a reviewer, I believe a major limitation of this work is its heavy reliance on hand-crafted heuristic features, which may lead to insufficient robustness when the system is confronted with diverse video distributions.

**Strengths And Weaknesses:**

\textbf{Strengths:}
-  \textbf{Well-Motivated Problem:} The paper accurately identifies and clearly articulates the ``delayed evidence'' and ``update pressure'' challenges in Streaming Video Understanding (SVU).
- \textbf{Creative Synthesis:} It cleverly integrates classic CV tracking heuristics (e.g., Mahalanobis distance) with LLM KV cache management, offering an inspiring, cross-disciplinary approach to token compression.
- \textbf{Practical Utility:} By providing a training-free, constant-memory-footprint solution, the method demonstrates high practical value for deploying vision-language models in resource-constrained streaming scenarios.


\textbf{Weaknesses:}

- \textbf{Severe Reproducibility Issues:} The mechanism relies heavily on complex heuristic rules governed by numerous hyperparameters (e.g., $W$, $K_{max}$, $S$, $\lambda_{sp}$, $\lambda_{idle}$, $T_{idle}$, EMA decay rates). However, the exact default values are entirely omitted from both the main text and the appendix. Furthermore, there is no sensitivity analysis. This makes the method impossible to reproduce and raises strong concerns about over-fitting to specific benchmarks.

-  \textbf{Incomplete Experimental Validation:}

       - \textbf{Unclear Baseline Configurations:} The exact detailed settings of the compared architectures are missing, which hinders fair comparison.
        - \textbf{Unclear Compression Ratio:} Raw input sequence lengths (total tokens before compression) are not reported for the delayed queries.
       - \textbf{Hidden Update Overhead:} Only query-time latency (TTFT) is evaluated. The potentially high online update/write overhead (calculating spatial distances, EMA, and residual statistics per frame) is ignored.


- \textbf{Missing Architectural Details:} The paper fails to explain how Positional Encodings (e.g., RoPE) are handled. It is unclear how position IDs are assigned to the synthesized far-memory ``pseudo-tokens'' to maintain temporal order when concatenated with near-window tokens.


The proposed mechanism relies heavily on spatial continuity and mass accumulation. I encourage the authors to discuss and provide supplementary analysis in the rebuttal on whether ProtoKV would experience catastrophic failures when facing frequent scene cuts, background dominance, and complex action dynamics.

---

> ### Author Rebuttal · Authors · 2026-03-29
>
> **W1. Reproducibility**
>
> We agree that the default hyperparameters should have been stated explicitly, and we now clarify them here. We use a fixed near–far allocation of $W : (K_{max} \cdot S) =1:3$ with $S=8$, and keep the remaining hyperparameters fixed at $(α, β, λ_{sp}, λ_{idle}, γ, T_{idle}) = (0.05, 0.05, 0.1, 0.01, 0.05, 120)$ and $(T_{maint}, ϵ_{K}, ϵ_{V}, n_{min}) = (1, 0.2, 0.25, 1)$. When $|M|$ changes, $W$ and $K_{max}$ scale accordingly under the same 1:3 split, while all other hyperparameters remain unchanged. We will open-source the implementation, configuration files, and exact experimental settings used in our evaluation.
>
> Sensitivity results are provided in our response to Reviewer 5VsY, W1: performance remains similar across balanced near–far allocations, with only the extreme near-heavy setting degrading at long delay, and 0.5×/2× coefficient sweeps produce only modest changes. Together with our use of the same defaults across all six benchmarks and tested memory settings without per-benchmark retuning, this alleviates concern that the reported gains arise from overfitting to specific benchmarks or domains.
>
> ---
> **W2. Experimental Validation.**
>
> *(1) Baseline configurations.*
>
> All methods are evaluated under the same benchmark protocol, video preprocessing pipeline, backbone, and total memory budget $|M|$, with the memory mechanism being the main difference. SWA uses the full $|M|$ as its sliding window, InfiniPot-V follows its original continual-compression setting ($α=0.5, γ=0.125$, far memory ratio $|C|/|M| = 0.75$; $|C|$: far memory size), and ProtoKV uses the same fixed near-far ratio of $W:K_{max}\cdot S = (1:3)$.
>
> *(2) Compression Ratio.*
>
> For delayed queries, the relevant pre-compression quantity is the raw input sequence length up to shifted query time $t_q$. For LLaVA-OV, streaming at 0.5 fps with 196 visual tokens per sampled frame yields about 98 tokens/sec, reaching up to 352.8k (RVS-Ego), 198.2k (RVS-Movie), and 205.6k (OVO-Bench), before compression to the fixed budget $|M|$. For Qwen2.5-VL, we follow the same model-side sampling configuration as InfiniPot-V, which caps the raw context at 49,920 tokens before compression to $|M|$. Thus, the effective compression factor varies with both query time and backbone.
>
> *(3) Update Overhead.*
>
> To measure the update overhead, we profile ProtoKV on Qwen2.5-VL-7B with RVS-Ego (RTX 5090). The ProtoKV update accounts for 33.8 ms/frame, i.e., about 30.8\% of the total per-frame processing time. This cost is incurred during stream ingestion rather than query-time answering, and in practical SVU pipelines with sampled frames and asynchronous queries, part of it may be absorbed into inter-frame gaps; it therefore does not directly affect query-time TTFT.
>
> ---
> **W3. Missing Architectural Details.**
>
> ProtoKV handles RoPE by assigning each prototype’s pseudo-tokens the same recency anchor $\tau_k$, preserving prototype-level temporal recency. Each far-memory prototype stores a recency anchor $\tau_k$, defined as the original position of the most recently absorbed source token assigned to prototype $k$. Thus, $\tau_k$ comes from the original stream positions, not from an external update clock. At query time, all pseudo-tokens synthesized from prototype $k$ receive RoPE using the same position $\tau_k$ before concatenation with the near-window KV cache. Near-window tokens keep their original positions, and prototypes are ordered by $\tau_k$.
>
> Using the average or first source-token position as $\tau_k$ performs worse than using the most recent one (59.6% and 57.2% vs. 61.0%, Qwen2.5-VL-7B, RVS-Ego), indicating that recency is the more effective positional anchor.
>
> ---
> **W4. Catastrophic Failure Analysis.**
>
> We evaluate three proxy tasks related to the reviewer’s concerns: Plot QA as the closest available proxy for scene-transition-heavy narrative videos, Anomaly Recognition for background-dominated content, and Ego Reasoning for complex ego-centric action dynamics, using Qwen-2-VL-7B under 6k budget.
>
> |MLVU task|Plot QA|Anomaly Recognition|Ego Reasoning|
> |-|-|-|-|
> |ProtoKV|65.7\%|69.5\%|61.9\%|
>
> These results do not suggest a distinct breakdown under frequent scene cuts, background-dominated content, or complex action dynamics. Plot QA is somewhat more challenging, but the pattern is not indicative of catastrophic failure. A clearer weakness is preserving boundaries among many visually similar repeated events over long horizons; see Reviewer 5VsY, W2.
>
> ---
> **Key questions**
>
> W1 and W4 address adaptability across varied scenarios: we use shared defaults across the six tested benchmarks/settings without per-benchmark retuning and we do not observe catastrophic failures in the additional proxy analyses. W2 clarifies the experimental comparisons, and W3 explains the RoPE handling for pseudo-tokens.
>
> ---
> We appreciate the reviewer’s feedback and will revise the paper accordingly.

---

### Official Review · Reviewer_4TWB · 2026-03-27

**Soundness:** 3
**Presentation:** 3
**Significance:** 3
**Originality:** 3
**Overall Recommendation:** 4
**Confidence:** 4

**Summary:**

This paper presents ProtoKV, a constant-footprint memory mechanism for streaming video understanding that targets the delayed-evidence challenge under bounded memory and latency. Across multiple SVU benchmarks, ProtoKV outperforms sliding-window and online token-retention baselines, with larger gains at longer delays.

**Compliance With Llm Reviewing Policy:**

Affirmed.

**Final Justification:**

The response resolves my concerns, and I will retain my positive score.

**Key Questions For Authors:**

Please refer to the Weaknesses section for the specific questions.

**Limitations:**

yes

**Strengths And Weaknesses:**

**Strengths:**

1. The paper is clearly motivated and identifies delayed evidence as an important challenge that SVU methods must address.

2. Representing far history as a fixed-capacity summary state is a reasonable design choice.

3. The paper is well written and easy to follow.

**Weaknesses:**

1. The evaluated backbones and baselines appear somewhat dated, so the generality of ProtoKV on newer foundation VLMs is unclear. Adding results on a more recent model such as Qwen3-VL would strengthen the claim that the method transfers well.

2. Compressing far-memory into prototypes may discard fine-grained evidence, so it is unclear how the method performs on queries requiring precise temporal dynamics or high-resolution spatial details.

3. The paper lacks concrete qualitative examples illustrating ProtoKV’s advantages. Adding a few specific scenario case studies would make the benefits more tangible.

4. The authors could include examples and a discussion of when ProtoKV fails and why.

---

> ### Author Rebuttal · Authors · 2026-03-29
>
> **W1. Backbone and Baseline Generality.**
>
> Evaluation on a newer foundation VLM such as Qwen3-VL would further strengthen the paper. At the same time, ProtoKV is a drop-in KV-cache-level method that does not modify model internals, and we already validate it across three distinct backbone families, LLaVA-OV-7B, Qwen2-VL-7B, and Qwen2.5-VL-7B, and four streaming benchmarks under the same default configuration rather than model-specific retuning.
> These consistent gains indicate that ProtoKV is not tied to one particular backbone architecture, while additional results on newer VLMs would further strengthen the generality claim.
>
> ---
> **W2. Fine-grained Evidence Loss.**
>
> Some loss of fine-grained evidence is an inherent trade-off under bounded far memory, since distant history cannot be preserved exactly once compressed into a fixed-capacity summary state. On the temporal side, we directly probe this issue using the predefined MLVU task types Action Order, Action Count, and Needle QA (see our response to Reviewer 5VsY, W2). These results show that ProtoKV is not limited to coarse event recall, although repeated-event counting remains challenging.
>
> On the spatial side, ProtoKV incorporates spatial information through its continuity-aware assignment rule: the spatial Mahalanobis term helps stabilize prototype association and reduce switching/fragmentation over time. This supports ProtoKV’s ability to maintain stable spatial association in far memory under a hard budget, complementing the temporal evidence above.
>
> ---
> **W3. Qualitative Examples.**
>
> Concrete qualitative examples help clarify this point, and representative success/failure cases are provided in our response to Reviewer 5VsY, W3. The analysis suggests that ProtoKV is more reliable for queries whose answers depend on localized state cues or relations among a small number of distinct events, because such evidence can remain stably represented in the summary-state far memory.
>
> In contrast, queries that require separating or counting many visually similar repeated events are more challenging, since these events can be absorbed into the same prototype and their boundaries become blurred over long horizons. This clarifies the trade-off more concretely.
>
> ---
> **W4. Failure Cases.**
>
> The clearest failure mode arises when ProtoKV must keep multiple visually similar events separated over long horizons under a fixed memory budget, especially for repeated-event counting queries. In this regime, several occurrences may be absorbed into the same prototype, so the far-memory summary retains cumulative evidence but not the boundaries between distinct episodes, which can lead to under-counting. This is consistent with our qualitative and task-level analysis: ProtoKV is less reliable when success depends on preserving repeated-event boundaries than when it depends on localized state cues or relations among a small number of distinct events. A representative example and related analysis are provided in our responses to Reviewer 5VsY, W2 and W3.
>
> ---
> We thank the reviewer and will reflect these points in the revision.

---

> > ### Author Rebuttal · Reviewer_4TWB · 2026-03-31
> >
> > Thank you for the rebuttal. The response resolves my concerns, and I will retain my positive score.

---

> > > ### Author Response · Authors · 2026-04-06
> > >
> > > Thank you for your thoughtful feedback. We are glad that our response addressed your concerns.

---

### Decision · Program_Chairs · 2026-04-30

**Decision:**

Accept (regular)

**Comment:**

All four reviewers weakly accept. ProtoKV offers a training-free, constant-footprint KV cache for streaming video understanding that effectively preserves delayed visual evidence via a two-tier memory architecture. The rebuttal has fully resolved all concerns; accept contingent on incorporating all rebuttal experiments into the final version.